# Metagenomics survey unravels diversity of biogas microbiomes with potential to enhance productivity in Kenya

**Samuel Mwangangi Muturi**[1,2]*, **Lucy Wangui Muthui**[3], **Paul Mwangi Njogu**[4], **Justus Mong'are Onguso**[2], **Francis Nyamu Wachira**[5], **Stephen Obol Opiyo**[6,7], **Roger Pelle**[3]

**1** Department of Biological Sciences, University of Eldoret, Eldoret, Kenya, **2** Institute for Bioteschnology Research, Jomo Kenyatta University of Agriculture and Technology, Juja, Kenya, **3** Biosciences Eastern and Central Africa—International Livestock Research Institute (BecA-ILRI) Hub, Nairobi, Kenya, **4** Institute for Energy and Environmental Technology, Jomo Kenyatta University of Agriculture and Technology, Juja, Kenya, **5** Department of Life Sciences, South Eastern Kenya University, Kitui, Kenya, **6** OARDC, Molecular and Cellular Imaging Center-Columbus, Ohio State University, Columbus, Ohio, United States of America, **7** The University of Sacread Heart, Gulu, Uganda

\* samuelmuturi37@gmail.com

**Data Availability Statement:** All relevant data are within the paper and its Supporting Information files.

## Abstract

The obstacle to optimal utilization of biogas technology is poor understanding of biogas microbiomes diversities over a wide geographical coverage. We performed random shotgun sequencing on twelve environmental samples. Randomized complete block design was utilized to assign the twelve treatments to four blocks, within eastern and central regions of Kenya. We obtained 42 million paired-end reads that were annotated against sixteen reference databases using two ENVO ontologies, prior to β-diversity studies. We identified 37 phyla, 65 classes and 132 orders. *Bacteria* dominated and comprised 28 phyla, 42 classes and 92 orders, conveying substrate's versatility in the treatments. Though, *Fungi* and *Archaea* comprised 5 phyla, the *Fungi* were richer; suggesting the importance of hydrolysis and fermentation in biogas production. High β-diversity within the taxa was largely linked to communities' metabolic capabilities. *Clostridiales* and *Bacteroidales*, the most prevalent guilds, metabolize organic macromolecules. The identified *Cytophagales*, *Alteromonadales*, *Flavobacteriales*, *Fusobacteriales*, *Deferribacterales*, *Elusimicrobiales*, *Chlamydiales*, *Synergistales* to mention but few, also catabolize macromolecules into smaller substrates to conserve energy. Furthermore, *δ-Proteobacteria*, *Gloeobacteria* and *Clostridia* affiliates syntrophically regulate $P_{H2}$ and reduce metal to provide reducing equivalents. *Methanomicrobiales* and other *Methanomicrobia* species were the most prevalence *Archaea*, converting formate, $CO_{2(g)}$, acetate and methylated substrates into $CH_{4(g)}$. *Thermococci*, *Thermoplasmata* and *Thermoprotei* were among the sulfur and other metal reducing *Archaea* that contributed to redox balancing and other metabolism within treatments. Eukaryotes, mainly fungi were the least abundant guild, comprising largely *Ascomycota* and *Basidiomycota* species. *Chytridiomycetes*, *Blastocladiomycetes* and *Mortierellomycetes* were among the rare species, suggesting their metabolic and substrates limitations. Generally, we observed that environmental and treatment perturbations influenced communities' abundance, β-diversity and reactor performance largely through stochastic effect.

**Funding:** This study was supported by BecA-ILRI-hub through the Africa Biosciences Challenge Fund (ABCF) fellowship program (13/RF/2017/4722) and University of Eldoret's annual research grant (UoE/B/DVASA/REG/97). The ABCF Program was funded by the Australian Department for Foreign Affairs and Trade (DFAT) through the BecA-CSIRO partnership; the Syngenta Foundation for Sustainable Agriculture (SFSA); the Bill & Melinda Gates Foundation (BMGF); and the UK Department for International Development (DFID) and; the Swedish International Development Cooperation Agency (Sida). The two forms of funds were awarded to SMM. The funders of ABCF fellowship program had no role in study design, data collection and analysis, decision to publish, or preparation of the manuscript.

**Competing interests:** The authors have declared that no competing interests exist.

Understanding diversity of biogas microbiomes over wide environmental variables and its' productivity provided insights into better management strategies that ameliorate biochemical limitations to effective biogas production.

## Introduction

The optimal utilization of micro-organisms is a critical strategy in the development of new products and improvement of the existing bioprocesses [1]. One of the widely applied technologies that utilises micro-organisms is anaerobic digestion (AD) of waste materials. The process generates biogas and digestate from wastes, the former being an energy carrier molecule [2]. Over 30,000 industrial installations globally, have been revealed to utilize AD to convert organic wastes into power, producing up to10,000 MW [3]. Apart from power production, stable AD processes provide one of the most efficient environmental models for organic waste valorization that enables nutrient recovery from the digestate [4], and hence providing additional economic benefits [5], when compared to other methods of waste management [6]. Many African countries, including Kenya produce huge amounts of waste [5] with high biogas generating potential. In spite of the efforts made to exploit this resource through the Biogas Partnership Programme [7], the biogas systems in African countries remain fairly rudimentary. The Programmes have installed approximately 60,000 biogas plants in Africa which include 16,419 in Kenya, 13,584 in Ethiopia, and 13,037; 7518, and 6504 in Tanzania, Burkina Faso and Uganda, repectively [5, 7]. Though Kenya leads in biogas exploitation in Africa, adoption of the technology remains poor, and is not commensurate with the country's energy needs. However, because of the relative popularity of the technology in the country, compared to its' counterparts, Kenya was justified as the appropriate pilot site for biogas studies among other African countries. Similar guidelines were also followed to select the Kenyan regions in this study.

Overall, it is thought that the poor adoption of the biogas technology in Africa is partly due to poor knowledge on the microbiomes that mediate the AD process. These processes comprise thermodynamic unfavorable reactions that contribute to stability and performance of the treatments. The microbial populations and the reactions involved are highy sensitive to environmental and reactor operation conditions; traits that prompt the need for development of sustainable management strategies for effective biogas production. Without such strategies, a complete breakdown of the whole system occurs. In the event of complete failure, farmers are forced to abandon the systems or invest in costly restoration processes, all of which end up reducing the economic and environmental benefits of the AD systems; thereby constraining efforts towards greening the economy.

Although, previous reports from other continents have indicated dominance of *Bacterial* and *Archaeal* species in tolerating reactor perturbations, limitations of traditional culture-dependent and independent methods e.g 16S rRNA gene [8, 9], still limit the knowledge on global biogas microbiomes diversities within these systems. The 16S rRNA techniques used thus far provide low resolution [10–12], and are not robust enough to unravel the high level of microbial diversity in complex biogas ecosystems. All these limit the information available on microbial composition and diversity which would be critical for improvement and better management of biogas systems. The 16S rRNA gene does not reveal the full extent of microbial diversity in any ecosystem [13–17] and most current diversity estimates are extrapolations of empirical scaling laws [13], and other theoretical models of biodiversity [14]. The above limitation constrains the capacity to scrutinize and apply the biodiversity scaling laws,

macroecological theories and other ecological theories [12–14] in the context of biogas production. This ends up directly limiting the available microbial diversity knowledge which would be useful to develop viable mitigition strategies to address biochemical reaction failures. Despite covering a small fraction of microbial diversity in a dataset, marker gene based studies, annotate the detected species against biological databases that comprise sequences of interest. This approach has the potential to lead to conflicts on the links between microbial populations and reactor performance. Such conflicts were reported by Wittebolle *et al.* [18] and Konopka *et al.* [19]. Wittebolle *et al.* considered higher microbial diversity as a reservoir of redundant metabolic pathways that possess desirable traits that respond to reactor pertubations [18] and other environmental conditions. In contrast, Konopka *et al.* argued that less diverse communities confer system stability by expressing complementary pathways [19] that avoid direct competition over the available resources. These kinds of contrasting views prompted us to ask whether microbial diversity in biogas reactors matters or it is the existence of the core microbiomes that determine the AD stability and maximum biogas production. In an effort to seek answers to these questions, we were provoked to further characterize the biogas system's micro-organisms using next generation sequencing platforms also termed "massively parallel sequencing" over a wide geographic area. Specifically, the technology employed sequencing-by-synthesis chemistry to generate the dataset. Previously, this technology has enabled massive sequencing of environmental genomes with high quality and accurracy [20]. However, the application of the technology in the biogas systems is still at its infancy. Furthermore, eukaryotes, mainly *Fungi* have normally been neglected by the majority of the studies, yet these species, are also crucial in the metabolism of large macromolecules in the AD systems. Our study was therefore designed to address the limitations enumerated above. We expect our findings to contribute to the formation of testable hypotheses for future application of metacommunity theory on the links between microbal diversity and environmental conditions. Further the study will contribute to a greater understanding of biogas microbiomes for improvement of biogas production.

## Results

### Annotations against the 16 biological databases

We used Illumina based shotgun metagenomics to identify and compare microbial communities within and among the biogas reactor treatments. In total 12 sludge samples assigned to twelve different treatments operating at a steady-state were assayed. The procedure yielded a total of 44 million raw reads and out of these, 2 million reads were of poor quality. We revealed 24,189,921 reads of 35-151bps from the remaining 42 million reads, which were filtered based on the % G:C content bias and duplicates according to Gomez-Alvarez *et al.* [21]. Approximately 23,373,513 reads conveying 1,261833 to 3,986675 reads per the treatment were realized (S1A Fig). Thereafter, we trimmed and assembled the contigs using *de novo* approach and the obtained high quality scaffolds (3,819,927 scaffolds, 1,119,465,586 bps, with mean average length of 291bps, and mean average standard deviation of 410 bps, S1 Table) were annotated against sixteen databases using two ENVO ontologies. Taxonomic assignments conveyed by SEED database were the highest (180, 000 scaffolds), while the assignment of NOG, LSU, RDP and COG databases were the least (<8000 scaffolds). Among the databases, IMG, Genbank and TrEMBL conveyed equal scaffold annotations (144,000 scaffolds). Other databases annotated the reads as indicated in Fig 1. Between the two ontologies, the large lake biome revealed the highest annotation compared to the terrestrial biome with a precision of *P≤0.05* across the treatments. Furthermore, 39.75% to 46.70% and 53.07% to 60.11% genes were conveyed to known and unknown proteins.

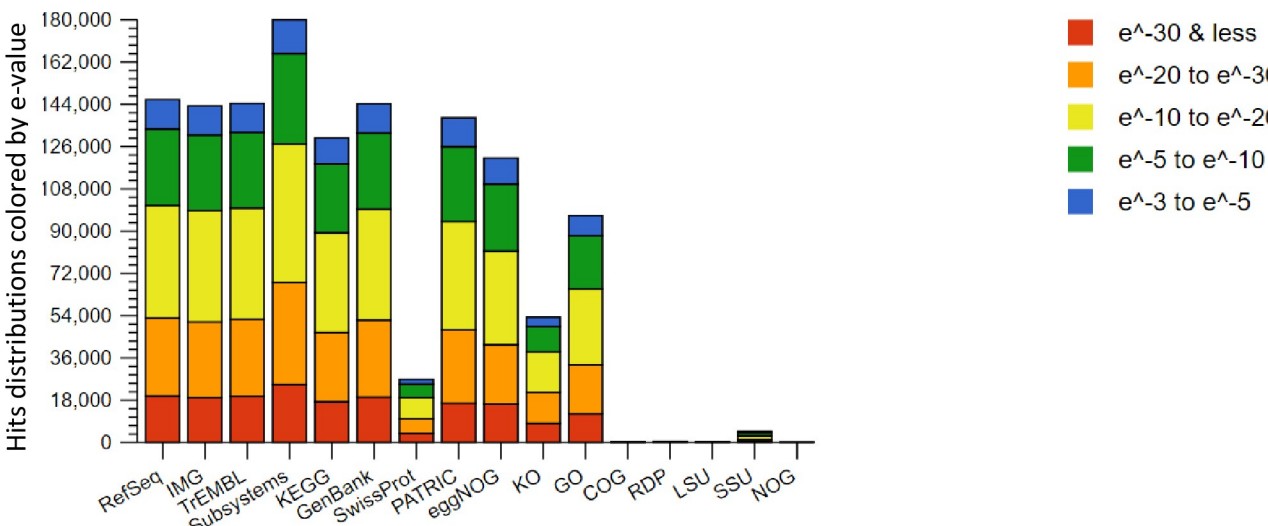

**Fig 1. Stacked barchart showing the sources of hits distributions from the sixteen databases.** These include nucleotide and protein databases, protein databases with functional hierarchy information, and ribosomal RNA databases. The bars representing annotated reads are colored by e-value range. Different databases have different numbers of hits but can also have different types of annotation data.

### Annotations against the archived taxonomic profiles

Taxonomic assignments revealed four broad but relatively simple distributions of microorganisms, with the species of three domains being responsible for methanogenesis. In summary, 2,617,791 reads hits were generated, with 213681 hits being assigned to *Archaea*, 2379293 hits to *Bacteria* and 21886 hits to *Eukaryotes* (S1A Fig). Thus, the revealed abundance hits ratio of *Bacteria* to *Archaea* to *Eukaryotes* in the reactor treatments was 109:9:1. The higher abundance of *Bacterial* species was linked to their metabolic capabilities. Unlike other cellular organisms, *Eukaryotes* mainly *Fungi* had the least abundances (< 1% of the total hits) among the treatments. Despite these variations of species abundances, we observed consistent microbial richness among the studied treatments. In total 37 phyla, 73 classes and 132 orders were indentified. A majority of the reactor treatments exhibited high community diversity, except for the communities detected in reactor 1, 3 and 6 that were found to cluster partially on the upper right quadrant of the plot (Fig 2), at a *P ≤ 0.05*. All the identified phylotypes were further annotated and β-diversity determined in details.

### The identified *Bacterial* biomes

We evaluated the variation in abundance and composition of 92 *Bacterial* domain orders that were assigned to 41 classes and 21 phyla (S2A Fig, Table 1). Generally we observed significantly higher *Bacterial* abundances (89–93% of the total hits), compared to other domains. Similarly, significantly high β-diversities of the *Bacteria* communities were also observed at the phylum level. However, the communities of reactor 2 and 10 (within the same block) and reactor 4 and 9 (on different blocks) were found to reveal low β-diversity (S2B Fig). In addition, we further observed low diversity variabilities between communities of reactor 3 and 6 and those of reactor 8 and 11 when compared at lower (class and order) taxonomic ranks (S3A and S3B Fig). Hence, only four out of twelve treatments had high β-diversities, which could be indicating genetic plasticity due to species evolution [22, 23].

Specifically, we revealed *δ-Proteobacteria* species as the most abundant guild, constituting 9–14% of the total microbial communities. Globally the species were the second most

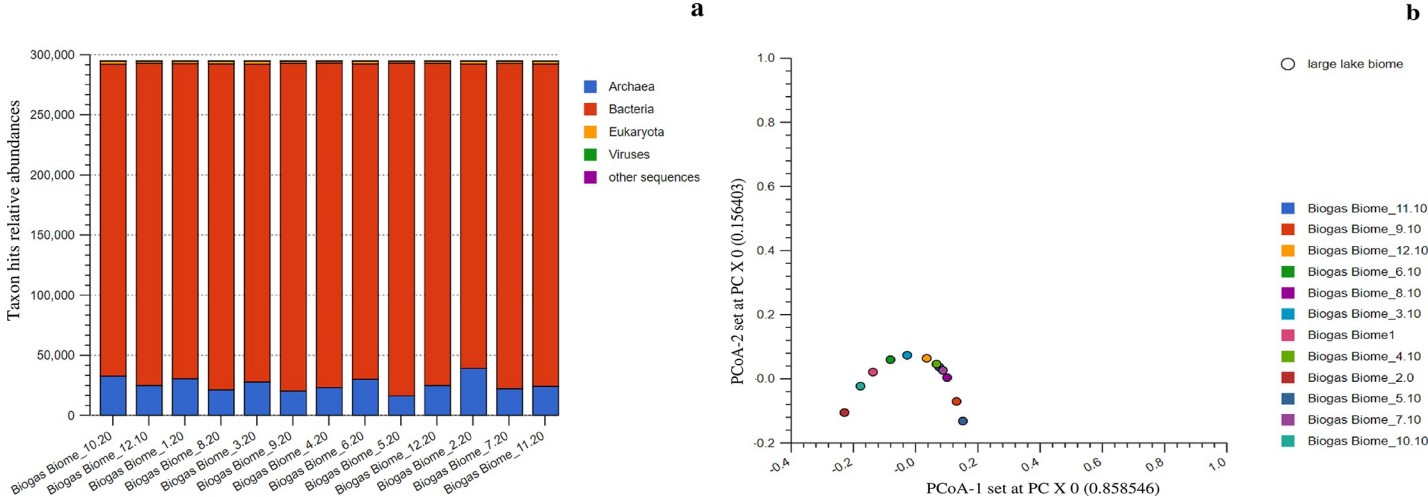

**Fig 2.** The stacked barchart (a) showing the three main domains, proportion of relative abundances and their PCoA plot (b) based on the Euclidean model. The PCoA plot revealed dissimilarities among the studied reactors. However, only three reactors partially clustered in the upper right quadrant of the plot. Though the Viruses nucleotide contributed to the curve they were not considered in the downstream analysis.

abundant after the *Clostridial* species that composed 11–26% of the total communities. The *δ-Proteobacteria* communities were classified into 7 orders (S6A Fig). Among them were *Syntrophobacteriales* that comprised 2–4% of the total identified communities, followed by *Desulfuromonadales* and *Myxococcales* species that comprised 2–4% and 0.9–4% of the total microbial compositions respectively. *Desulfobacterales* and *Desulfovibronales* communities were found to comprise 1–2% of the total species abundances. Among the least abundant *δ-Proteobacterial* species were the affiliates of *Desulforculales* and *Bdellovibrionales* that comprised only < 1% of the identified total communities. The non-classified reads derived from the class *δ-Proteobacteria*, also composed < 1% of the total communities. However, the PCoA plots indicated high sample to sample variation of the *δ-Proteobacteria* communities, although the composition of reactor 4 and 7 were found to be similar (S6B Fig).

Among the identified species, the affiliates of *Firmicutes* were the most abundant in seven out of twelve reactor treatments and comprised 15–34% of total hits (Table 1) while the *Proteobacteria* species (Fig 3) dominated in the other five treatments, comprising of 19–32% of total hits (Table 1). *Proteobacteria* richness was significantly higher compared to that of *Firmicutes* species. The *Proteobacteria* communities were classified into 38 orders, which were affiliated to 7 classes (S4A Fig). We linked their versatility with their ability to circumvent the oxidative stress in the systems. Significantly high levels of similarity of microbial composition were revealed among the treatments 1, 3 and 6, and treatments 4, 7 and 12, all located on the upper and lower right quadrant of the plot, (S4B and S5 Figs; at the class and order level). However, low β-diversities were observed for the identified *Proteobacteria* communities of reactor 4, 7, 8 and 9, (S5 Fig) at the order level. Other treatments were found to have high β-diversities of *Proteobacteria* species.

The communities' affilliated to the class *γ-Proteobacteria* comprised 4–11% of the total identified reads. The affiliates were the fifth most abundant group in the entire microbiome and were classified into 14 orders (S7A Fig). Among the identified orders was *Pseudomonadales*, that dominated the class, comprising of <1 to 6% of the total identified communities. This phylotype was followed by *Alteromonadales* and *Enterobacterales* species in abundance. The members of *Chromatiales*, *Vibrionales*, *Methylococcales*, *Oceanspirillales* and *Pasteurelales* respectively were also among the detected species. Other identified orders were

**Table 1. The Krona software generated *Bacterial* phyla biomes of the twelve treatments within eastern and central regions of Kenya.**

| Phylum | S_11.10 | S_9.10 | S_12.10 | S_6.10 | S_8.10 | S_3.10 | S_1 | S_4.10 | S_2.0 | S_5.10 | S_7.10 | S_10.10 |
|---|---|---|---|---|---|---|---|---|---|---|---|---|
| *Firmicutes* | 27% | 34% | 26% | 21% | 25% | 22% | 22% | 31% | 15% | 30% | 29% | 15% |
| *Proteobacteria* | 26% | 19% | 20% | 27% | 24% | 27% | 25% | 19% | 31% | 25% | 21% | 32% |
| *Bacteroidetes* | 15% | 18% | 22% | 11% | 18% | 11% | 15% | 20% | 10% | 16% | 18% | 10% |
| *Actinobacteria* | 4% | 3% | 5% | 5% | 4% | 4% | 4% | 4% | 7% | 5% | 7% | 6% |
| *Chloroflexi* | 3% | 2% | 3% | 6% | 4% | 6% | 4% | 3% | 7% | 2% | 3% | 7% |
| *Spirochaetes* | 2% | 3% | 1% | 3% | 2% | 3% | 4% | 2% | 2% | 1% | 1% | 1% |
| *Verrucomicrobia* | 2% | 2% | 1% | 2% | 1% | 2% | 2% | 1% | 2% | 0.9% | 1% | 2% |
| *Cyanobacteria* | 2% | 1% | 2% | 2% | 2% | 2% | 2% | 2% | 3% | 1% | 2% | 3% |
| *Planctomycetes* | 2% | 1% | 2% | 3% | 2% | 3% | 3% | 2% | 3% | 2% | 2% | 3% |
| *Chlorobi* | 1% | 1% | 1% | 1% | 2% | 1% | 1% | 1% | 1% | 0.9% | 1% | 1% |
| *Acidobacteria* | 1% | 1% | 1% | 2% | 1% | 2% | 1% | 0.9% | 2% | 0.8% | 1% | 2% |
| *Thermotogae* | 1% | 0.9% | 1% | 1% | 1% | 1% | 1% | 0.9% | 1% | 0.8% | 1% | 1% |
| unclassified reads | 0.6% | 0.6% | 1% | 1% | 2% | 1% | 0.9% | 1% | 0.6% | 1% | 1% | 0.7% |
| *Synergistetes* | 0.8% | 0.6% | 1% | 0.7% | 0.7% | 0.6% | 0.6% | 0.8% | 0.8% | 0.6% | 0.8% | 0.9% |
| *Lentisphaerae* | 0.8% | 0.5% | 0.3% | 0.4% | 0.3% | 0.4% | 0.4% | 0.3% | 0.3% | 0.2% | 0.4% | 0.2% |
| *Fusobacteria* | 0.6% | 0.8% | 0.6% | 0.5% | 0.6% | 0.5% | 0.5% | 0.7% | 0.4% | 0.7% | 0.7% | 0.4% |
| *D. Thermus* | 0.5% | 0.4% | 0.6% | 0.9% | 0.6% | 0.8% | 0.7% | 0.6% | 1% | 0.5% | 0.6% | 1% |
| *Aquificae* | 0.5% | 0.4% | 0.4% | 0.5% | 0.5% | 0.5% | 0.5% | 0.4% | 0.4% | 0.3% | 0.4% | 0.5% |
| *Tenericutes* | 0.3% | 0.6% | 0.3% | 0.2% | 0.4% | 0.3% | 0.2% | 0.3% | 0.2% | 1% | 0.5% | 0.3% |
| *Deferribacteres* | 0.3% | 0.3% | 0.3% | 0.4% | 0.4% | 0.4% | 0.3% | 0.3% | 0.3% | 0.2% | 0.3% | 0.3% |
| *Nitrospirae* | 0.3% | 0.2% | 0.3% | 0.4% | 0.4% | 0.4% | 0.4% | 0.3% | 0.4% | 0.2% | 0.3% | 0.4% |
| *Chlamydiae* | 0.3% | 0.2% | 0.1% | 0.2% | 0.2% | 0.1% | 0.2% | 0.1% | 0.2%% | 0.08% | 0.2% | 0.1% |
| *Fibrobacteres* | 0.2% | 0.5% | 0.1% | 0.2% | 0.1% | 0.2% | 0.2% | 0.2% | 0.08% | 0.2% | 0.2% | 0.08% |
| *Dictyoglomi* | 0.2% | 0.2% | 0.2% | 0.4% | 0.3% | 0.4% | 0.3% | 0.3% | 0.3% | 0.2% | 0.2% | 0.3% |
| *Elusimicrobia* | 0.2% | 0.2% | 0.1% | 0.1% | 0.1% | 0.2% | 0.1% | 0.1% | 0.1% | 0.1% | 0.2% | 0.1% |
| *Chrysiogenetes* | 0.08% | 0.06% | 0.07% | 0.07% | 0.07% | 0.09% | 0.06% | 0.07% | 0.09% | 0.05% | 0.07% | 0.09% |
| *Gemmatimonadete* | 0.06% | 0.06% | 0.07% | 0.1% | 0.1% | 0.1% | 0.1% | 0.07% | 0.2% | 0.6% | 0.08% | 0.2% |
| *C. Poribacteria* | 0.03% | 0.03% | 0.03% | 0.05% | 0.03% | 0.05% | 0.03% | 0.03% | 0.05% | 0.02% | 0.03% | 0.04% |

**S_1:** Biogas Biome1; **S_2.0:** Biogas Biome 2.0; **S_3.10:** Biogas Biome 3.10; **S_4.10:** Biogas Biome 4.10; **S_5.10:** Biogas Biome 5.10; **S_6.10:** Biogas Biome 6.10; **S_7.10:** Biogas Biome 7.10; **S_8.10:** Biogas Biome 8.10; **S_9.10:** Biogas Biome 9.10; **S_10.10:** Biogas Biome 10.10; **S_11.10:** Biogas Biome 11.10; and **S_12.10:** Biogas Biome 12.10.

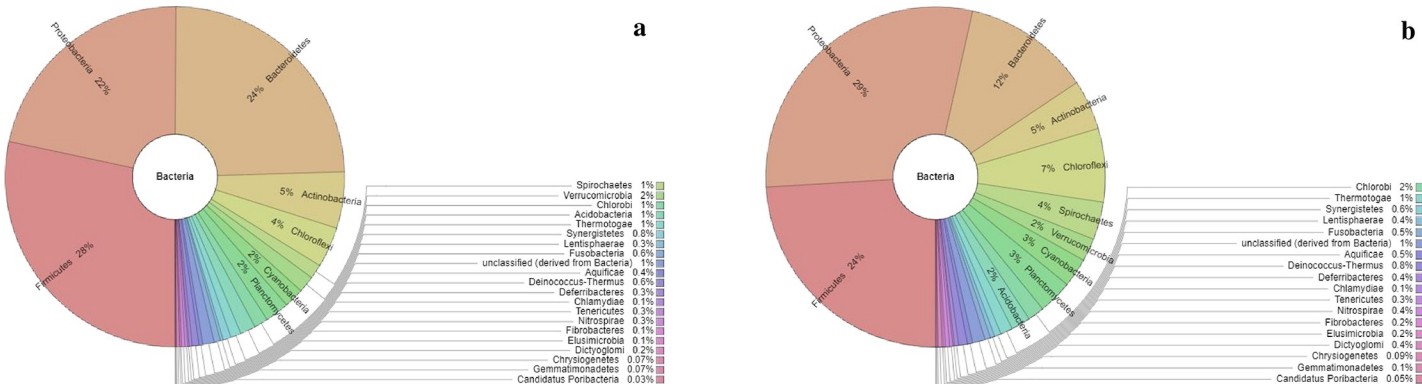

**Fig 3. The Krona radial space filling display showing the source evidences of the stated relative abundance of the identified communities in the text (e.g Table 3 and within the text).** The display reveals the abundances of the Bacterial reads for the communities identified in reactor 12 and 3: a) Reveals community abundances of reactor 12 while b) reveals community abundances of reactor 3, where *Firmicutes* and *Proteobacteria* dominated.

**Table 2. The geographic positioning systems' co-ordinate for the sampled reactor treatments, assigned to the four blocks of eastern and central regions of Kenya.**

| Reactor Number | Sample id | Blocks | Treatment | Northings | Eastings | Altitude | Location |
|---|---|---|---|---|---|---|---|
| EA12/11/09/004 | S_1 | Kiambu | L-C | S(00˚15.477') | E (37˚38.828') | 1468 M | Ngenda |
| EA12/10/07/003 | S_2 | Kiambu | S-C | S(00.21347˚) | E(37.61867˚) | 1674 M | Karimoni |
| EA12/10/04/001 | S_3 | Meru | S-E | S(00˚08.180') | E(37˚39.477') | 1443 M | Kaithe |
| CE20/12/01/020 | S_4 | Kirinyaga | S-C | S(00˚32'45.97") | E(37˚26'40.52") | 1364 M | Murinduko |
| EA12/10/08/004 | S_5 | Meru | L-E | N(00˚01.997') | E(37˚35.353') | 1958 M | Kithaku |
| EA12/10/09/002 | S_6 | Meru | M-E | N(00˚04.046') | E(37˚38458') | 1650 M | Ntima |
| EA12/11/12/002 | S_7 | Meru | L-E | N(00.02946˚) | E(37.10077˚) | 1987 M | Ontulili |
| EA14/11/05/001 | S_8 | Embu | L-E | S(00˚033.269') | E(37˚28.247') | 1306 M | Municipality |
| EA14/11/09/001 | S_9 | Embu | M-E | S(00˚27.824') | E (37˚32.537') | 1417 M | Gaturi South |
| CE12/10/06/012 | S_10 | Kiambu | L-C | S(00.24874˚) | E(37.62438˚) | 1589 M | Kianyau |
| CE12/10/07/013 | S_11 | Kiambu | S-C | S(00˚16'17.57") | E(37˚38'40.16") | 1577 M | Dururumo |
| CE14/11/11/001 | S_12 | Kirinyaga | M-C | S(00.48637˚), | E (37.44057˚) | 1539 M | Ngandori |

**CE**: Central regions, **EA**: Eastern regions, **L-C**-Large size, Central regions, M-C: Medium size, Central region, **S-C**: Small size, Central regions, **L-E**: Large size, Eastern regions, M-E: Medium size, Eastern regions **S-E**: Small size, Eastern regions.

*Xanthomonadales, Thiotrichales, Legionellales, Aeromonadales, Acidithiobacillales* and *Cardiobacterales*, all comprising <1% of the total identified communities. Unclassified *γ-Proteobacteria* reads were also identified in this study. Our results also revealed significantly high β-diversity ($P \leq 0.05$) of the microbial communities in the following six reactor treatments (reactor 2, 5, 6, 10, 11, 12). In contrast, the communities identified in reactors 1, 3, 4, 7, 8 and 9 were found to cluster on the upper right quadrant of the PCoA plot (S7B Fig), indicating homogeneity and genetic composition similarity within the microbial populations.

Communities of *β-Proteobacteria* were detected in low abundances (2–4% of the identified microbial communities). Affiliates of *Burkholderiales* comprised of 1–2% of the microbial populations while the affiliates of *Rhodocyclales, Neisseriales,* and *Nitrosomonadales* species were found to comprise of <1% of the total identified populations across the treatments. Other identified *β-Proteobacteria* species were assigned to the order *Methylophilales, Gallionellales* and *Hydrogenophilales* which comprised of <1% of the total identified communities in the twelve treatments. High sample to sample variation was also observed for the *β-Proteobacteria* species (S8B Fig).

The affiliates of *α-Proteobacteria* were revealed to comprise of 3–8% of the total microbial communities, and were classified into 7 orders (S9A Fig). The *Rhizobiales* dominated this group, comprising of 1–2% of the total microbial population in the treatments. The relative abundances of *Rhodobacterales, Rhodospirillales,* and *Sphingomonadales* followed (representing <1% of the total population). Other *α-Proteobacteria* orders identified included *Caulobacterales, Rickettsiales* and *Parvularculales.* We also found significantly high dissimilarities among the *α-Proteobacteria* within the studied treatments. Only the *α-Proteobacteria* communities of reactor 3 and 11 were revealed to express low β-diversity (S9B Fig), a trait that we postulate was attributed by similar reactor operating conditions rather than the observed environmental conditions (Tables 2 and 3).

The ε and Z-*Proteobacteria* were among the least abundant *Proteobacteria* phylotypes in our study and were found to comprise of ≤1% of the total identified microbial populations in the treatments. The *ε-Proteobacteria* communities were classified into the orders *Campylobacteriales* and *Nautiliales* (S10A Fig), both comprising of <1% of the total identified populations. We found similarities between the *ε-Proteobacteria* communities of reactor 4 and 8 (clustered)

**Table 3. The ArcGIS software and the questionnaire generated environmental variables and phenotypic traits for the sampled treatments within the identified blocks of Kenya.**

| Sam_id | Alt | S-T | AEZ | AAR | AMT ℃ | Substrate source | | | Family size | | | Reactor Performance | | | | |
|--------|-----|-----|-----|-----|-------|----|----|----|---|---|---|----|-----|----|-------|-----|
| | | | | | | CO | PO | PI | M | F | T | IG | ACG | RV | FACoG | GGV |
| S_1 | 1468 M | Lithosol | UM 4 | 800–900 | 20.7–19.9 | 4 | - | - | 5 | 4 | 9 | 4.8 | 4.0 | 16 M$^3$ | 14 M$^3$ | 2 |
| S_2 | 1674 M | Ferralsol | UM 2 | 1100–1400 | 19.5–18.4 | 3 | - | - | 2 | 1 | 3 | 2.4 | 2.0 | 8 M$^3$ | 7 M$^3$ | 1 |
| S_3 | 1443 M | Nitosol | UM 3 | 1400–2200 | 20.6–19.2 | 11 | - | - | 3 | 4 | 7 | 2.4 | 2.0 | 8 M$^3$ | 5 M$^3$ | 3 |
| S_4 | 1364 M | Andosol | UM 2 | 1220–1500 | 20.1–19.0 | 6 | - | - | 4 | 2 | 6 | 1.8 | 1.5 | 6 M$^3$ | 4.5 M$^3$ | 1.5 |
| S_5 | 1958 M | Phaeozem | L H 1 | 1700–2600 | 17.4–14.9 | 3 | - | - | 3 | 1 | 4 | 4.8 | 4.0 | 16 M$^3$ | 12 M$^3$ | 4 |
| S_6 | 1650 M | Nitosol | UM1 | 1650–2400 | 19.2–17.6 | 2 | 30 | - | 2 | 2 | 4 | 3.0 | 2.5 | 10 M$^3$ | 6 M$^3$ | 4 |
| S_7 | 1987 M | Phaeozem | LH 1 | 1700–2600 | 17.4–14.9 | 5 | - | - | 5 | 2 | 7 | 4.8 | 4.0 | 16 M$^3$ | 10 M$^3$ | 6 |
| S_8 | 1306 M | Nitosol | UM 3 | 1000–1250 | 20.7–19.6 | 5 | - | 6 | 5 | 4 | 9 | 4.8 | 4.0 | 16 M$^3$ | 11 M$^3$ | 5 |
| S_9 | 1417 M | Andosol | UM 2 | 1200–1500 | 20.1–18.9 | 2 | - | - | 3 | 2 | 5 | 3.0 | 2.5 | 10 M$^3$ | 8 M$^3$ | 2 |
| S_10 | 1589 M | Acrisols | UM 2 | 1100–1400 | 19.5–18.4 | 3 | - | - | 2 | 2 | 4 | 4.8 | 4.0 | 16 M$^3$ | 13 M$^3$ | 3 |
| S_11 | 1577 M | Cambisol | UM 3 | 800–1200 | 19.9–19.5 | 14 | - | - | 5 | 4 | 9 | 2.4 | 2.0 | 8 M$^3$ | 6 M$^3$ | 2 |
| S_12 | 1539 M | Ferralsol | UM1 | 1400–1700 | 19.3–17.5 | 3 | - | - | 4 | 3 | 7 | 3.0 | 2.5 | 10 M$^3$ | 8 M$^3$ | 2 |

**Sam_id:** Sample id; **Alt:** Altitude (M a.m.s.l); **S-T**: Soil type; **AEZ**: Agro-ecological zones; **UM 3**: Marginal Coffee zone; **UM 1**: Coffee-Tea Zone; **UM 2**: Main coffee zone; **LH 1**: Tea-Dairy Zone; **AAR**: Annual Average rainfall (mm); **AMT**: Annual mean Temperature ($^0$C); **CO**: Cow; **PO**: Poultry; **PI**: Pigs; **M**: Male; **F**: Female; **T**: Total; **IG**: Ideal gas (M$^3$); **ACG**: Actual calculated gas (M$^3$); **RV**: Reactor Volume (M$^3$); **FACoG**: Farmers Actual Collected gas (M$^3$); **DGV:** Difference between the RV and FACoG.

and those of reactor 5 and 9 (clustered partially) revealing low β-diversities (S10B Fig). All the identified *Z-Proteobacteria* were from the order *Mariprofundales*, and accounted for <1% of the total microbial populations in the studied treatments.

The *Firmicutes* communities were the second most diverse community identified in this study. They comprised of 15–34% of the total microbial communities identified. The community was classified into 8 orders and 4 classes (S11A and S12A Figs). The *Firmicutes* in the three treatments of reactor 2, 10 and 12 revealed significant genetic similarity and were revealed to cluster in the lower left quadrant of the PCoA plot (S11B Fig). The *Firmicutes* in reactor 1 and 7 were found to cluster partially on the upper right quadrant of the PCoA plot (significant threshold; *P≤0.05*; S11B Fig), indicating low β-diversity. At the order level, the identified *Firmicutes* of reactor 2 and 10; 3 and 6; 7 and 12; and reactor 1 and 9 also exhibited low β-diversities (S12B Fig). Other local communities were revealed to express high β-diversities at class and order levels.

The *Clostridia* was revealed to be one of the most abundant *Firmicutes* phylotype, comprising 8–22% of the identified total microbial communities. It was classified into 4 orders (S13A Fig). The order *Clostridiales* dominanted the phylum and constituted 8–22% of the total communities. *Thermoanaerobiales* species constituted 2–3% of the total identified communities. Other identified *Clostridia* species were the affiliates of *Halanaerobiales* and *Natranaerobiales*, all representing <1% of the total microbial populations, except for the *Halanaerobiales* communities in reactor 11 that comprised of 2% of the total identified microbial populations. Among the twelve biogas reactor treatments, three treatments (i.e reactor 7, 11 and 12) clustered on the lower left quadrant of the PCoA plot (S13B Fig), indicating high community similarities. Communities of reactor 5 and 9 were found to cluster partially on the upper left quandrant of the PCoA plot, revealing significantly low β-diversity. The other seven treatments had high β-diversities, implying metabolic and genetic plasticity [22, 23] among the identified *Clostridia* species. The class *Bacilli* comprised of 4–9% of the total identified communities and was the fourth most abundant *Clostridia*. The *Bacilli* were classified into *Bacillales* and

*Lactobacillales* (S14A Fig), which comprised of 3–4% and 1–4% of the total identified micro-biome communities, respectively. Among the *Bacilli* communities, those of reactor 5 and 8 were found to exhibit significant dissimilarity, while the other treatments exhibited low β-diversities (S14B Fig). Other detected low abundance *Firmicutes* were *Erysipelotrichales* (*Erysi-pelotrichi*) and *Selenomonadales* (*Negativicutes*) which comprised of <1% of the total identified communities. The *Erysipelotrichales* species in reactor 5 and 9 were however found to com-prise of 1% of the total identified microbial populations.

*Bacteroidete*s were the third most abundant phylotype comprising of 10–22% of the total identified microbial populations. These were classified into 4 classes and orders (S15A and S16A Figs). Among them, were the *Bacteroidia* populations that comprised of 5–12% of the identified microbial communities. All the identified *Bacteroidia* species belonged to the order *Bacteroidales*. Other identified members of *Bacteroidete*s were *Flavobacteria* that belonged to the order *Flavobacterales*. The communities of *Cytophagia* were all assigned to *Cytophagales* species and they comprised of 1–2% of the total identified microbial communities. The other *Bacteroidete*s were *Sphingobacteria*, mainly of the order *Sphingobacteriales*, which comprised of 2–3% of the total identified microbial communities. We observed partial clustering of the *Bacteroidetes*' communities particularly in reactor 2 and 10 and in reactor 8 and 12, which indicated low β-diversity between the respective communities (S15B and S16B Figs). The *Bac-teroidete*s communities of reactor 4 and 9 also clustered on the lower left quadrant of the PCoA plot (S15B and S16B Figs), implying significant similarities of the *Bacteroidete*s in the two treatments.

We identified 6 orders of *Actinobacteria* species (S17A Fig) which comprised of 3–7% of the identified microbial reactor communities. The *Actinomycetales*' were among the identified affiliates, comprising of 2–6% of the total identified communities. The species of order *Corio-bacteriales* and *Bifidobacteriales* accounted for <1% of the total identified microbial composi-tion. Other less abundant *Actinobacteria* species were assigned to the order *Rubrobacteriales*, *Solirubrobacteriales* and *Acidimicrobiales*. Out of the twelve biogas reactor treatments, the microbial communities of the seven treatments (reactor 1, 3 and 12; reactor 5 and 11 and reac-tor 7 and 8; S17B Fig) were found to exhibit significantly ($P{\leq}0.05$) low β-diversities. Only *Acti-nobacteria species* of the remaining three out of the twelve treatments were found to exhibit high β-diversities.

The *Chloroflexi* were categorized into 4 classes (S18A Fig) and 5 orders (S19A Fig) and comprised of 2–7% of the total microbial communities identified (Table 1). The communities belonged to *Chloroflexi* (1–4% of the identified communities) and *Dehalococcoidetes* (${\leq}$1–3% of the identified communities) as the most abundant phylotypes. Although all the *Dehalococ-coidetes* reads were unclassified at a lower level, we identified *Chloroflexales* and *Herpetosipho-nales* as the affiliates of the class *Chloroflexi* (S20 Fig). The two orders comprised of 1–2% and <1% of the total identified microbiomes, respectively. Other identified classes of the phylum were *Thermomicrobia* and *Ktedonobacteria*; all of which comprised of <1% of the total identi-fied communities. The *Thermomicrobia* were classified into orders *Sphaerobacterales* and *Thermomicrobiales* (S21A Fig). All the identified *Ktedonobacteria* belonged to the order *Ktedo-noacteriales* (also representing <1% of the identified communities). We identified two treat-ments (reactor 1 and 7) that revealed low β-diversity of their local *Chloroflexi* communities (S18B Fig). Four treatments (reactor 4, 5, 10 and 12), out of the remaining ten were revealed to also cluster on the lower right quandrant of the PCoA plot (S18B and S18B Fig).

Similarly, the *Chloroflexi* communities of reactors 1 and 5 and those of reactors 3 and 7 were found to cluster on the upper right and lower right quandrants of the PCoA plot (S20B Fig). The *Chloroflexi* class communities of reactor 1 and 10 were found to cluster partially with the cluster of microbes from reactor 3 and 7. We also identified partial clustering of *Chloroflexi*

from reactor 8, 11 and 12 indicating low β-diversities among the communities in these reactors (S20B Fig). The *Chloroflexi* from the rest of the treatments revealed high β-diversities. In contrast, the *Thermomicrobia* communities were significantly dissimilar among the treatments as indicated in S21B Fig.

The *Cyanobacteria* comprised of 1–3% of the total microbial composition and majority in this group were unclassified. Only <1% of the total population were assigned to the class *Gloeobacteria* (S22A Fig). Unlike other phylotypes, only the *Cyanobacteria* communities of reactors 3, 7 and 10 were found to exhibit significantly high β-diversities ($P \leq 0.05$), (S22B Fig) while the communities in the other treatments were found to express partial similarities between or among themselves. The *Gloeobacteria* communities were all classified as *Gloeobacteriales* order. Other identified communities (unclassified reads) included members of the following orders; *Chroococcales*, *Nostocales*, *Oscillatoriales* and *Prochlorales* (S23A Fig). Overall, majority of the local *Cyanobacteria* communities were found to reveal low β-diversities. The *Cyanobacteria* communities in reactors 4, 6, 9 and 12, however exhibited high β-diversities (S23B Fig). Similarly the identified four phylotypes at the order level (derived from unclassified reads, S24A Fig) were also revealed to express high local compositional dissimilarities among the twelve biogas treatments (S24B Fig).

We also identified *Acidobacteria* which comprised of 1–2% of the total identified microbial communities. Majority of these populations were from the order *Acidobacteriales* (comprising <1–2% of the total identified microbial communities). The remaining species were affiliated to *Solibacteres*, particularly of the order *Solibacterales* (<1% of the total identified microbial communities) (S25A and S26A Figs). The *Acidobacteria* communities of five (reactors 1, 2, 3, 8, 12) out of twelve treatments were revealed to exhibit low β-diversities. *Acidobacteria* communities of reactor 2 and 8 clustered on the upper left quadrant of the PCoA plot (at order level; S25B and S26B Figs), indicating similarity between the two local communities. *Deinococcus-Thermus*, another phylum, had the orders *Deinococcales* and *Thermales* species (S27A Fig), the two belonging to the class *Deinococci*. The relative abundances of the two orders were ≤1% of the total identified microbial populations. We observed low β-divesity between the *Deinococci* species of reactor 11 and 12 while the diversities of its communities in reactors 3 and 6 and those of reactors 7 and 9 were found to be significantly similar (S27B Fig). All other local *Deinococcus-Thermus* communities expressed high β-diversities.

The identified *Verrucomicrobia* communities were classified into 3 classes that comprised of 1–2% of the total identified microbial communities. Among the identified classes were *Opitutae*, *Verrucomicrobiae* and *Spartobacteria* (S28A Fig). The affiliates of *Verrucomicrobiales* were the most abundant followed by those of the *Puniceicoccales* and *methylacidiphilales* species; all representing <1% of the total identified communities (S29B Fig). Notably, we found out that all the *Opitutae* and *Spartobacteria* reads were unclassified at a lower taxonomic level. Only the *Verrucomicrobia* communities of reactors 3 and 7 were found to cluster partially on the lower right quadrant of the PCoA plot, (at the class level, S28B Fig). In contrast, several local communities including the ones identified in reactors 1 and 10, and reactors 3, 6, 7 and 11 were found to cluster partially (indicative of low β-divesities) within the plots (at order level; S29B Fig). However, their communities in reactor 5 and 8 were found to cluster significantly on the upper right quadrant of the PCoA plot (S29B Fig), implying high similarity between the *Verrucomicrobia* species in the two treatments. Other treatments were found to reveal high β-diversities. Among the identified *Tenericutes*, were *Acholeplasmatales*, *Mycoplasmatales* and *Entomoplasmatales*, all comprising of <1% of the total identified microbial communities and belonged to the class *Mollicutes* (S30A Fig). Interesting, member species of this class were obsolutely absent in reactor 5 only, contrary to other phyla species, probably due to natural selection pressure. Our data also revealed significantly high similarities of the identified

*Tenericutes* species inhabiting reactors 4 and 9 and reactors 6 and 8 (S30B Fig). The local communities of the other treatments were observed to be significantly dissimilar.

Our results also identified *Planctomycetacia* affiliates which comprised of 1–3% of the total identified microbial communities. These affiliates were all assigned to the order *Plantomyceta-cetales* (*Planctomycetacia*). *Chlorobia* were identified in our study (comprising of 1–2% of the total identified communities). These belonged to the order *Chlorobiales*. *Spirochaetes* were also identified (<1–4% of the total identified microbial composition). All the *Spirochaetes* identified in this study belonged to the order *Spirochaetales*. *Thermotogae* of the order *Thermotogales* were also identified in the study. Other less abundant communities identified were *Synergistales* (*Synergistates*, *Synergistia*), *Lentisphaerales* (*Lentisphaerae*, *Lentisphaerae*), *Fusobacterales* (*FusoBacteria*, *FusoBacteria*), and *Aquificales* (*Aquificae*, *Aquificae*), all of which comprised of <1% of the total identified communities. *DeferriBacteriales*, *Chlamydiales*, *Chrysiogenales*, *Nitrospirales*, *Fibrobacteriales*, *Elusimicrobiales*, *Dictyoglomales*, *Gemmatimonadales* were other groups of microbes identified in this study. The *Candidatus poribacteria* affiliates were not accurately classified at the lower taxonomic ranks (Table 1).

## The identified *Archaeal* biomes

We analyzed the identified *Archaea* domain and established that their species comprised of 6–12% of the total microbial populations. Out of the identified communities, we obtained 5 phyla (S31A Fig), 9 classes (S32A Fig) and 16 orders (S33A Fig). Among the communities, those identified in reactor 4 and 12 were found to cluster on the lower left quadrant of the PCoA plot, indicating their high similarity. We also revealed low β-diversities of the identified *Archaeal* communities, of reactor 3 and 6 and also those detected in reactor 7 and 9 (Phylum level; S31B Fig). However, at a lower taxonomic level, only those identified in reactor 3 and 7 were observed to have low β-diversity (Class and order level; S32B and S33B Figs). All the other reactor treatments were observed to contain highly dissimilar communities of *Archaea* at different taxonomic ranks.

A majority of *Archaea* species identified in this study were classified as *Euryarchaeota* and comprised of 5–11% of the total microbial communities. The *Euryarchaeota* communities were classified into 8 classes (S34A Fig) and 10 orders (S35A Fig). Interestingly, we observed significant dissimilarities (high β-diversities) of *Euryarchaeotes* among the reactor treatments at the two taxonomic levels of class and order (S34B and S35B Figs). The class *Methanomicrobia* was the most abundant *Euryarchaeota* phylotype and comprised of 4–9% of the total identified microbial populations. Further, we established that the majority of the members of the class were affiliates of *Methanomicrobiales* order (representing 2–4% of the identified total communities), followed by those assigned to *Methanosarcinales* and *Methanocellales* (S36A Fig), which comprised of 1–3% and ≤1% of the identified total microbial communities, respectively. The local composition of *Methanomicrobia* species were highly dissimilar among the reactor treatments except for those from reactor 1 and 7 that were found to cluster. The communities of the two treatments revealed significant similarity (significant threshold, $P \leq 0.05$; S36B Fig). Other identified methanogenic communities were assigned to *Methanobacterales*, the class *Methanobacteria*; *Methanococcales* belonging to *Methanococci* class; and *Methanopyrales* the affiliates of *Methanopyri* phylotype. All these species were revealed to comprise of <1% of the detected microbial communities.

Other non-methanogenic *Euryarchaeotes* identified in our study were the affiliates of *Thermococcales* (*Thermococci*), *Archaeglobales* (*Archaeglobus*), *Halobacterales* (*Halobacteria*), and *Thermoplasmatales* (*Thermoplasmata*). Among these phylotypes of *Euryarchaeotes*, affiliates of *Thermococcales* were dominant while the *Thermoplasmatales* were the least abundant.

*Crenarchaeota* and *Thaurmarchaeota* phyla were among the other identified non-methano-genic species. Between the two phyla, *Crenarchaeotes* were the most diverse and were classified into 4 orders (S37A Fig). Within the identified *Crenarchaeota*, the following orders were iden-tified; *Desulfurococcales*, *Thermoproteales*, *Sulfolobales* and *Acidilobales* (in order of abun-dance). All these orders are members of *Thermoprotei* class which comprised of <1% of the identified microbial communities. We noted that half of the studied treatments (reactor 6 and 9, reactor 7 and 10 and reactor 2 and 11) had *Crenarchaeota* communities that had low β-diversities, while the rest of the communities, in the other six treatments had significantly high β-diversities (significant threshold, $P \leq 0.05$; S37B Fig). *Thaurmarchaeota*, another non-metha-nogenic phylum was classified into *Nitrosopumilales* and *Cenarchaeales* orders (S38A Fig) both of which exhibited high β-diversities (S38B Fig). Other identified phyla were *Korarch-aeota* and *Nanoarchaeota*. The classification of these two phyla at the lower taxonomic ranks was inaccurate and the sequence reads from the two were therefore treated as unclassified.

## The identified *Fungal* biomes

Our study also identified the presence of *Fungal* species. The *Fungal* communities were classi-fied into 5 phyla (S39A Fig), 14 classes (S40A Fig) and 23 orders (S41A Fig). Their relative abundances were however low and they comprised of <1% of the total microbial populations among the reactor treatments. Significantly high β-diversities were noted for the identified local communities of *Fungi* at the phyla and class levels (S39B and S40B Figs). However, at the lower taxonomic levels, low β-diversities were the common feature particularly for the com-munities identified in reactor 2 and 10, reactor 3 and 6, reactor 4 and 9, and reactor 8 and 11 (S41B Fig).

The affiliates of *Ascomycota* were the most abundant guild (representing <1 of the total communities) within the *Eukaryotes* and were classified into 7 *Ascomycota* classes (S42A Fig) that included *Eurotiomycetes*, *Saccharomycetes* and *Sordariomycetes*, respectively. Other identi-fied classes were *Schizosaccharomycetes*, *Dothideomycetes*, *Leotiomycetes* and *Pezizomycetes* all of which comprised of <1% of the total microbial communities in the respective treatments. We also further classified the identified *Ascomycota* species into 11 orders (S43A Fig). Interest-ingly we observed significantly low β-diversity between *Ascomycota* communities in reactor 7 and 8. Other identified *Ascomycota* communities were observed to exhibit high β-diversities at the lower taxonomic levels (class and order level; S42B and S43B Figs). Among the classes, *Sor-dariomycetes* were the richest within the phylum and their species were classified into 4 orders; the *Hypocreales*, *Sordariales*, *Magnaporthales* and *Phyllachorales* (S44A Fig). The *Sordariomy-cetes'* communities were found to exhibit significantly high β-diversities (Statistical threshold, $P \leq 0.05$) (S44B Fig) among the studied treatments. Despite their higher abundance compared to *Sordariomycetes* phylotypes, the *Eurotiomycetes'* another class of *Ascomycota* were classified into two orders; *Eurotiales* and *Onygenales* (S45A Fig). The two *Eurotiomycetes* orders were shown to comprise of <1% of the identified total microbial populations. Like other fungal phy-lotypes, only the *Eurotiomycetes'* communities of reactor 1 and 3 were found to cluster on the PCoA plot, which was indicative of high community similarity. The other reactor communities were revealed to have significantly dissimilar *Eurotiomycetes* populations (significant thresh-old, $P \leq 0.05$; S45B Fig). Other identified fungal species were the affiliates of *Saccharomycetales*, *Schizosaccharomycetales*, *Pleosporales*, *Helotiales* and *Pezizales* phylotypes, all comprising <1% of the total microbial communities.

Among the identified *Fungi*, *Basidiomycota* species were the second most abundant (repre-senting <1% of the total communties). The species were classified into 4 classes and 5 orders (S46A and S47A Figs, respectively). Among the classes, *Agaricomycetes* dominated the

treatments, followed by *Tremellomycetes*, *Ustilaginomycetes*, and *Exobasidiomycetes* species (S46A Fig) respectively. The affiliates of the class *Agaricomycetes* were further classified as *Agaricules* and *Polyporales* species (S48A Fig). Other *Basidiomycota* species were affiliated to the following orders; *Tremellales* (*Tremellomycetes*), *Ustilaginales* (*Ustilaginomycetes*) and *Malasseziales* (*Exobasidiomycetes*). We observed two treatments (reactor 11 and 12) that contained *Agaricomycete*'s communities with low β-diversity (S48B Fig) while the local *Basidiomycota* communities in the other reactor treatments had significant, high β-diversities (S46B and S47B Figs).

The *Chytridiomycotes* were among the identified rare species. This was classified into *Chytridiomycetes* (more abundant) and *Monoblepharidomycetes* (S49 Fig). Among the *Chytridiomycetes* were *Spizellomycetales* (identified in reactor 9 and 10), *Rhizophydiales* and *Cladochytriales*, the latter two dectected only in reactor 9 (S50 Fig). All the affiliates of *Monoblepharidomycetes* were classified as *Monoblepharidales* species and were also detected in reactor 9 only. Other rare phylotypes identified in our study included *Blastocladiales*, (*Blastocladiomycetes*, *Blastocladiomycota*, detected in reactor 9), *Entomophthorales*, *Mortierellales*, and *Mucorales*, all derived from unclassified *Fungal* reads (also identified in reactor 9, 10 and 11 respectively, S51 Fig). *Microsporidia* species were not accurately classified at the lower taxonomic levels and were regarded as unclassifed though they comprised of <1% of the total microbial populations.

## The microbiomes α-diversity and biogas production

Since environmental and technical variables can outweigh the biological variations when handling metagenomic datasets, prior to analysis, we asked ourselves, whether we had attained the maximum microbial richness and if our sampling was a representative of this maximum. To answer these questions, we conducted a rarefaction analysis through statistical resampling and plotted the total number of distinct species annotations against the total number of sequences sampled (Fig 4). As a result, a steep slope, that leveled off towards an asymptote was obtained for each sample and later combined (Fig 4). This finding indicated that a reasonable number of individual species were sampled and more intensive sampling of the same treatments would yield probably few additional species, if not none. Further the α-diversity indices revealed 947–1116 genera across the treatments, comfirming our maximum sampling effort was attained for further diversity studies. The study also revealed variabilities of reactor performance i.e biogas productivity (Table 3) among treatments, the phenotype that we linked to the observed α-diversity variability, among the sites.

## The microbiomes β-diversity and biogas production

To understand the influence of environmental variations and other variables on the local microbial communities and thus biogas production, we conducted comparative community diversity studies on the twelve biogas reactor treatments. As a result, we revealed high β-diversity among the local communities in the majority of the treatments (at the domain level). The only observed exceptions were the local communities of reactor 4 and 11 that clustered on the lower left quadrant of the plot (S52A Fig), revealing significant similarity between the organisms in the two treatments. Similarly, at a lower taxa level (phylum, class and order), only the communities of reactor 2 and 10 were found to be similar, and were located in either the lower right or upper left quadrants of the PCoA plot (S52B and S52C Fig). We also observed low β-diversity between the communities of reactor 4 and 9 at the lower taxonomic levels (phylum, class and order) with species clustering partially on the lower left and upper right quadrants of the PCoA (S52B and S52C Fig), respectively. Other treatments that revealed low β-diversity

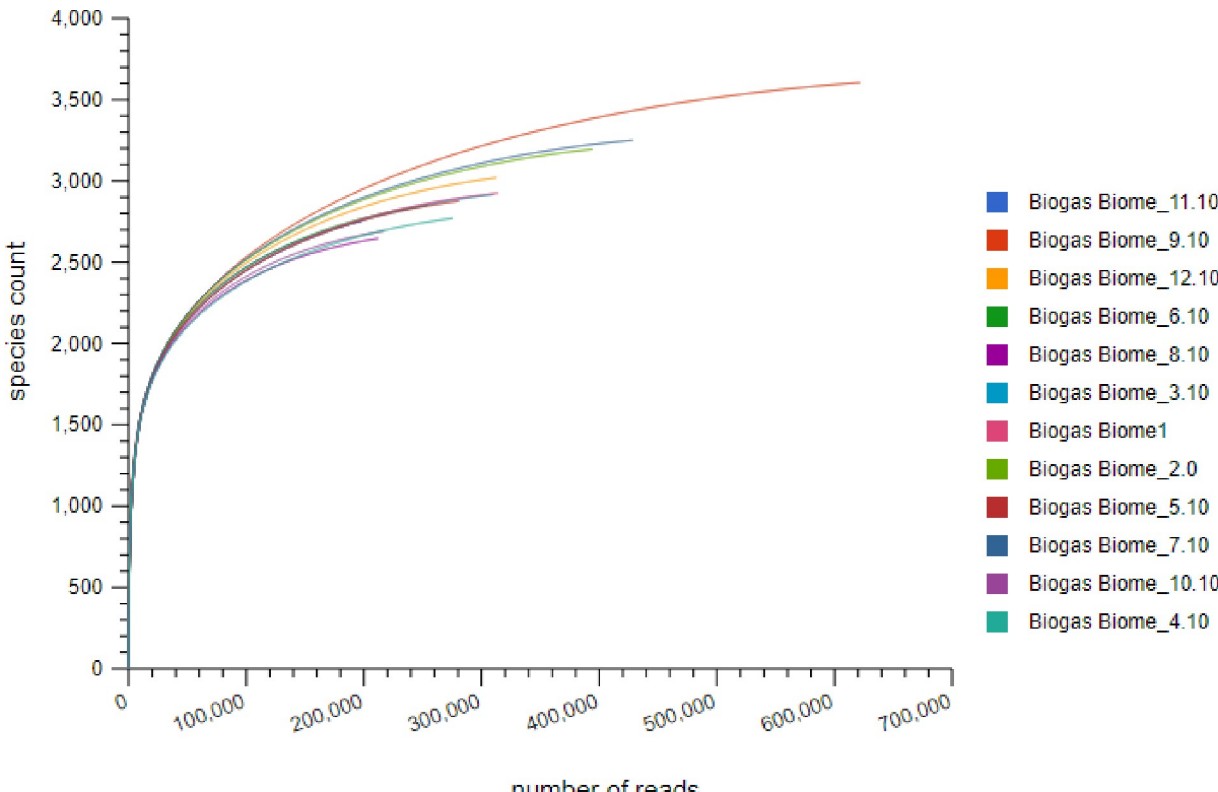

**Fig 4. Rarefaction plot showing a curve of annotated species richness.** The curve is a plot of the total number of distinct species annotations as a function of the number of sequences sampled.

between their communities were reactor 8 and 11 and reactor 3 and 6. Communities from these reactors were found to cluster partially in the lower right and lower left quadrants of the plot (Fig 5). Similarly, we also noted significant variability of biogas production among the treatments (Table 3), an indication of the observed β-diversities at different taxonomic levels and within the taxa (stated above).

## Discussion

### Annotations against the 16 biological databases

The aim of our study was to use next generation sequencing technologies and different bioinformatics tools to investigate the biogas microbiomes from environmental samples obtained from a wide geographic coverage. We combined marker genes and other genes on the assembled scaffolds to generate maximum taxonomic assignments. Among the sixteen utilized databases, SEED subsystems, yielded the highest read hits annotations, the findings that corroborated with the previous reports [24]. Unlike other databases, SEED integrated several evidences that provided annotation benefits over the other 15 utilized databases. PATRIC database had fewer annotations compared to the IMG database, due to its high speciality of storing only pathosysstems information. The annotation of low protein sequences by the SWISS-PROT database was due to its manual curation and annotation systems. Contrasting the findings were the annotations of TrEMBL and Refseq databases that both utilized automated schemes, which superseded the annotation of SWISS-PROT database. However, in spite of its low resolution, majorly due to lack of regular updates, the outputs of SWISS-PROT

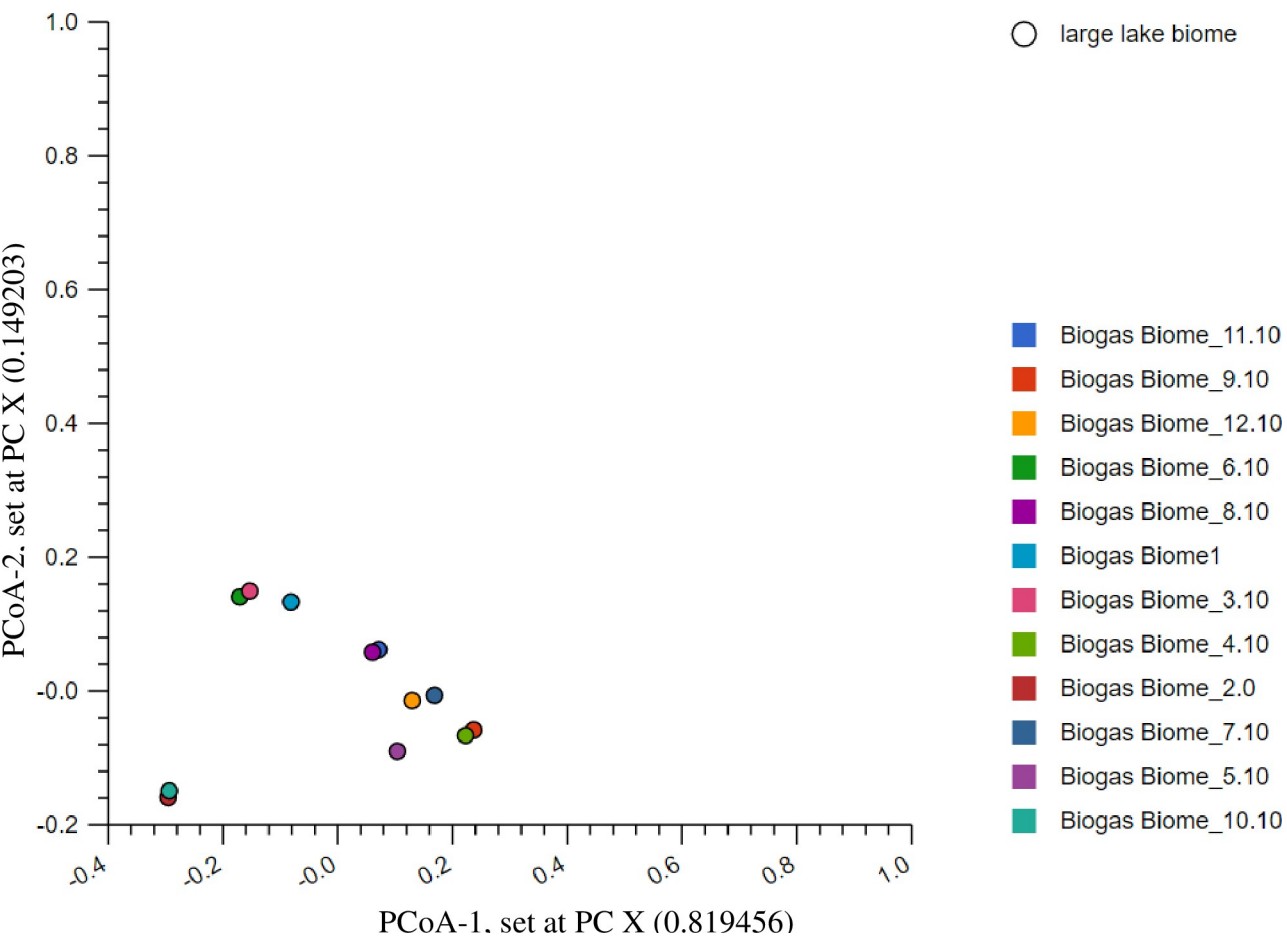

**Fig 5.  The PCoA analysis revealing β-diversities of the twelve treatments at the order level.** The reads compositions in reactor 2 and 10 were partially similar, located in the lower left quadrant, those in reactor 8 and 11 located in the upper right quadrant, reactor 4 and 9 in the lower right quadrant, while those in reactor 3 and 6 were located in the lower left quadrant of the plot.

are more accurate compared to the automated databases. Furthermore, orthologous groups databases such as COG, eggNOG etc were utilized to distigush orthologs from paralogs. In this case, we coupled gene classification schemes to the taxonomic assignements in order to increase detection resolution. Among these five databases (Fig 1), eggNOG had the highest hits annotation due to its high phylogenetic resolution, automated annotation scheme, and coverage of more genes and genomes than its counterpart databases. In contrast, COG and NOG databases were among the databases that had the lowest hits annotations because of their manual curation and annotation schemes and fewer regular updates due to manual labor requirements. KEGG database was used to reconstruct metabolic pathways of the identified taxonomic assignements. Other utilized nucleotide databases were Genbank that ensured uniform and comprehensive annotation against the worldwide nucleotide sequences, Ribosomal Database Project (RDP) database that covered small rRNA gene subunit (SSU) of *Archaea* and *Bacteria* domains, LSU and SSU databases that archived large and small rRNA subunit of *Archaea*, *Bacteria* and *Eukarya* sequences respectively. The latter two; provided a complete species detection advantage over the RDP database. The two databases archived nucleotide sequences together with their postulated secondary structures. The SSU database stored more community sequences than its counterpart; the LSU database and it used prokaryotic and

eukaryotic models for secondary structure alignments. The extra feature of the SSU database explained why the database, had the highest annotation resolution compared to other amplicon sequence databases. However, it's worth-noting that the three rRNA databases archived complete or near-complete rRNA sequences, although partial fragments of 70% of the estimated chain length of the combined molecule were also included. These characteristics of the amplicon sequence databases limited the annotation of our Miseq generated short reads. The ENVO ontologies analysis, which revealed high resolution of large lake biomes, suggested that majority of the cultured and sequenced species were derived from large water masses.

## Annotations against the archived taxonomic profiles

Our taxonomic annotations approach generated more detection, compared to the other annotations conducted elsewhere [25]. While other studies revealed 0.042% to 0.132% rRNA genes, we revealed 280 (0.00%) to 1,032 (0.00%) rRNA genes. These findings implied that the proportion of the rRNA genes to the total shotgun generated genes were negligible. The observations were highly attributed to our sampling and sequencing approaches as previously explained by Knight *et al.* [26]. Furthermore, while most of the studies used 19–35% of the total generated sequences [25], our approach annotated and used 39.75–46.70% of our total generated sequences. All these findings indicate that more than 53% of the biogas generating populations is still unknown. Briefly, our annotations were assigned to *Bacteria*, *Archaea*, *Eukaryotes* and *Viruses*, which corroborated the findings of other studies [27]. Among the domains, *Bacterial* species dominated in abundance, comprising 87–93% of the total microbial populations, followed by *Archaea* and *Eukaryotes*, while *Viruses* comprised the least abundant population. The observed high *Bacterial* abundances, suggest their crucial metabolic roles in biomass conversion and other reactions within the reactor systems [27]. As expected, we found low abundance among the *Eukaryotes* (*Fungi*) which was indicative of the presence of low recalcitract biomasses concentration. Generally, the *Fungal* species are involved in several biochemical reactions of biogas production including hydrolysis, fermentation and syntrophic reactions. The identified affiliates of *Archaea* were majorly consumers of smaller substrates that were generated by the *Bacterial* and *Fungal* species. The *Archaea* species are able to use different methanogenic routes to convert the substrates into $CH_{4(g)}$. Nevertheless, the main roles of the identified and less abundant group of the virus were unclear, though the species could have been active in degrading other microbial cells in the AD systems. However, their detection was largely attributed to the viral pathogenicity on both *Bacteria* and *Archaea* species. High β-diversities among the local communities (at the domain level) were an indication of environmental influences at the different ecological location.

## The identified *Bacterial* biomes

More than 90 representative orders of the *Bacteria* domain were revealed in our datasets. The first step of the AD process involves hydrolytic reactions that convert large macromolecules into smaller substrates [28]. Several *Bacterial* communities are capable of hydrolysis, even on high lignocellulosic plant biomasses. The identified species that participate in these reactions were annotated to the *Firmicutes*, *Bacteroidetes*, *Chloroflexi* to mention but a few and their species are known to harbor cellulosomes, that secrete extracellular enzymes, to degrade the substrate. Previous reports have indicated that species belonging to *Bacteroidales*, *Bifidobacteriales*, *Clostridiales Lactobacillales*, *Thermotogales* and many other identified *Bacterial* species, catalyze the first step of AD [28]. The step involves rate limiting reactions, largely regulated by the substrate composition. The nature of the chemical composition of the substrate, generally depends on the environmental variables (Tables 2 and 3), which influence

microbial species abundances and β-diversity differently. All these affects reactor stability and performance. As expected, the class *Clostridia*, that comprised *Clostridiales*, *Thermoanaerobacteriales* and *Halanaerobiales* orders, was the most dominant among the identified phylotypes. The guild converts large macromolecules into organic acids, alcohols, $CO_{2(g)}$ and $H_{2(g)}$ [29–35]. In this case, the species use a substrate-level phosphorylation mechanism to conserve the energy. Acetogenic and homoacetogenic species belonging to the same guild such as *Moorella thermoacetica* and *Thermoanaerobacter kivui* are able to use the reductive acetyl-CoA pathway to reduce $CO_{2(g)}$ into $CO_{(g)}$, concomitantly generating reducing equivalents [36, 37] and vice versa. As a consequence, the acetogens and homoacetogens use proton or sodium dependent mechanisms to conserve their energy [37]. Above all, the *Clostridia* guilds are able to catalyze other biochemical reactions such as reductive dechlorination of various compounds that leads to production of sulfide from either thiosulfate or sulfite. The reactions consume pyruvate and lactate as the carbon source. In this case, the member species are able to use [NiFe]-hydrogenases to generate $H_{2(g)}$ [35]. In our systems, we also identified *Halanaerobiales* species that have a capacity to tolerate high salt concentrations [38]. Generally the reactor treatments comprised high salts concentrations (treated cow dug) and the *Halanaerobiales* must have regulated the toxic cations from the ionized salts. All these roles of *Clostridia* suggest their metabolic versatility, and these features were used to explain the observed low richness of *Firmicutes* compared to *Proteobacteria* species. Similarly, the observed high β-diversities among the treatments were linked to the versatile roles of *Clostridia*. However, the observed low β-diversities (S13B Fig) of the *Clostridia* communities in reactor 7, 11 and 12 were attributed to similar treatment perturbations and the identified environmental variables (Tables 2 and 3).

The *Bacillales* and *Lactobacillales* species identified in our study, all belonged to the class *Bacilli*. These species are able to use mixed fermentation routes to metabolize carbohydrates and other biological macromolecules [39, 40]. Formate dehydrogenase complex activities are the main mechanisms of energy conservation [40] in these microbial guilds. Generally, *Bacilli* co-exist in symbiotic relationship with *Clostridia* species in lactate and acetate metabolism [41], a mechanism that is termed as lactate cross-feeding. High β-diversities among the observed treatments (S13B Fig) were solely attributed to cow-dung's substrate composition and environmental variables (Tables 2 and 3). However, the low β-diversities observed between or among the treatments (S14B Fig) that formed the clusters were largely attributed to similarities of reactor's operation conditions. All these implied that the respective organisms were in the same succession stage at the time of sampling. The *Erysipelotrichi* another identified *Firmicutes* class, utilize the syntrophic acetogenesis route to metabolize lipids into acetate, $H_{2(g)}$ and $CO_{2(g)}$ [25]. Its species, together with other *Firmicutes* species like *Smithella propionica* use non-randomizing pathway [42] to metabolize propionate into acetate. Among other identified *Firmicutes*, were *Syntrophomonas wolfei* (*Clostridiales*, *Clostridia*) that use β-oxidation pathway to oxidize $C_4$ to $C_7$ fatty acids in order to produce either acetate and $H_{2(g)}$ or propionate, acetate and $H_{2(g)}$ [43, 44]. Notably, these syntrophic species are consumers of 30% of the generated electrons, which bypasses the rate limiting steps of the volatile fatty acids accumulation. All these reactions, contributes to reactor stabilities. In general, we associated the observed low *Firmicutes* abundances and diversities (richness, S11A and S12A Figs) with genetic versitility that conferred capacity to encode an assortment of proteins; which metabolize a variety of biomasses into biogas. Furthermore, the observed high β-diversities (S11B and S12B Figs) of the *Firmicutes* support our concepts on the genetic divergence among the species and we postulate that the trait was mainly influenced by the size of the treatment and the variability of the environmental conditions (Tables 2 and 3).

The *Synergistete*s species identified in this study have been established to ferment amino acids into lactate, acetate, $H_{2(g)}$ and $CO_{2(g)}$ [45, 46]; though they have no ability to metabolize carbohydrates and fatty acids. One such identified example is *Dethiosulfovibrio peptidovorans*. The affiliates of genus *Anaerobaculum* which belong to *Synergistete*s catabolize peptides into acetate and short-chain fatty acids while their close relatives, *Aminobacterium*, are capable of fermenting amino acids into acetate, propionate and $H_{2(g)}$ [45, 46]. Other amino acid metabolizers identified in this study were the affiliates of *Fusobacteria* [39, 47] which are able to catabolize substrates into acetate, propionate, and butyrates, concomitantly releasing $H_{2(g)}$, $CO_{2(g)}$ and $NH_{3(g)}$. The member species of the two phyla are known to use substrate-level phosphorylation to conserve their energy [48] from amino acids metabolism. Our findings contradict the metabolism of the amino acid by the *Clostridia* species, whose energy conservation mechanisms remain unclear and have been postulated to occur through chemiosmosis [32]. However, the two phyla described here have been found to generate energy through syntrophic association with methanogens; otherwise the generated protons would increase the systems' alkalinity and lead to inhibited steady-state of the biochemical reactions. Based on the low abundance of the two phyla and richness among the treatments, we postulate low amino acid affinities among their identified species, which would suggest another selection pressure in the AD systems [32].

The identified affiliates of *Bacteroidales*, *Flavobacterales*, *Cytophagales* and *Sphigobacteriales*, all belong to *Bacteroidetes*, and are majorly known to ferment carbohydrates and proteins into mixed products (acetate, lactate, ethanol, and volatile organic acids), concomitantly releasing $H_{2(g)}$ [49]. Moreover, their species are also capable of metabolizing aliphatic, aromatic and chloronated compounds [50] and were observed to co-exist with methanogens possibly to increase energy extraction from indigestible plant materials. The member species were highly varied among the treatments and we hypothesize that the cattle breeds and the operating conditions in the treatments were the key drivers of the observed β-diversities. Interestingly, we also observed consistency in the relative abundance (2% of total reads) of the *Cytophagale*s guild among the sites, which was attributable to long periods of low nutrient limitation [51]. Although, the metabolic activity of the identified *Chlorobi* overlap with that of *Bacteroidetes* [52], their species use reverse tri-carboxylic acid (rTCA) cycle to fix $CO_{2(g)}$ and $N_{2(g)}$ in anaerobic conditions. We hypothesize that these rTCA reactions may have been utilized by the *Chlorobi* to oxidize reduced sulfur compounds [53, 54]. In our results, the *Bacteroidetes* species however revealed a higher selection advantage over *Chlorobi* due to the observed low *Chlorobi* abudunces and richness (i.e with only a single order, the *Chlorobale*).

The identified *Proteobacteria* guilds consisted of metabolically versatile species and were revealed to comprise the highest number of species (richness) compared to other identified phyla in the treatments. The guilds comprised of species with capacity to metabolize proteins [55] carbohydrates and lipids [56] and we associated their low β-diversity (S4B and S5B Figs) with low cow-dug's substrate composition heterogeneity and differences in environmental conditions (Tables 2 and 3). The *γ-Proteobacteria* was revealed in our study as the most diverse class of the *Proteobacteria* and the class comprised of 14 orders (S7 Fig). The *Enterobacteriales* included *Escherichia coli*, which are facultative anaerobes able to ferment carbohydrates into lactate, succinate, ethanol, acetate, $H_{2(g)}$ and $CO_{2(g)}$ [57]. Unlike other syntrophic species, the affiliates of *Enterobacteriales* have been found to use methyl-citrate cycle [58] to metabolize propionate into pyruvate, a precursor for butanol, isopropanol and other mixed acid fermentation products. *Burkholderiales*, another *γ-Proteobacteria* phylotype identified in this study, produces succinate from carbohydrates concomitantly oxidizing $H_{2(g)}$, $S^0$ and $Fe^0$ into their respective oxidized forms to conserve energy [59, 60]. We suggest that the observed high

abundances and β-diversities of the *γ-Proteobacteria* communities (S7 Fig, Table 3) could be due to low environmental temperature.

In contrast, the affiliates of genus *Sutterella*, among other species of *β-Proteobacteria* identified in this study are known saccharolytic and nitrate reducers [60]. The species also consume propionate, butyrate and acetate syntrophically [56, 60]. We suggest that the high abundance and diversity of *β-Proteobacteria* (S8 Fig) in the treatments was favoured by the agroecological zones and prevailing environmental temperatures (Tables 2 and 3) of the reactors. The *Caulobacterales*, an order of *α-Proteobacteria* were among the identified low abundant species in our study. These species have previously been revealed to survive in low nutrient environments [61]. *Rhodospirillales* (*α-Proteobacteria*), such as *Rhodospirillum rubrum*, have been previously determined to participate in homoacetogenesis and convert $CO_{(g)}$ and formate into acetyl-CoA, $CO_{2(g)}$ and $H_{2(g)}$ [62]. The produced acetyl-CoA is further converted into acetate, the precursor for methanogenesis. Like members of *β-Proteobacteria*, the *α-Proteobacteria* species identified in this study were also influenced by the agroeclogical zones (Tables 2 and 3) and we postulate that the cow-dug's substrate composition was the main cause of the observed β-diversities and abundance variations (S9 Fig) among the treatments. The *Campylobacterales* and *Nautiliales* species which are affiliates of *ε-Proteobacteria* were also identified in our study. These species metabolize proteins in the AD systems and have also been revealed to express an additional role of antibiotic resissitance in biofilms and granules [56].

Among the identified *δ-Proteobacteria* species, *Pseudomonadales*, the gemmules producing anaerobes [63] were the most abundant and have been found to co-exist with algae [63]. The reported syntrophic traits of the *Pseudomonadales* order were revealed to enable the species to withstand unfavourable environmental conditions, which could have offered the advantage to the species against the selection pressure; over other *δ-Proteobacteria* species. Generally, the *δ-Proteobacteria* species are syntrophic oxidizers [64–66], with the ability to convert volatile fatty acids into acetate, formate, $CO_{2(g)}$ and $H_{2(g)}$. The *Syntrophobacteriales* species identified in this study e.g *Syntrophus aciditrophus* are able to use randomized methyl-malonyl-CoA pathway to oxidize butyrate, propionate and long-chain fatty acids into acetate, formate, $CO_{2(g)}$ and $H_{2(g)}$ syntropically [64, 65] while *Syntrophobacter wolinii* [66] another member species, is known to use β-oxidation pathway to metabolize propionate into acetate. Other species belonging to the genus *Syntrophaceticus* identified in this study have been reported to oxidize the produced acetate into $CO_{2(g)}$ and $H_{2(g)}$ [67]. Interestingly, for these species to effeciently oxidize the respective macromolecules, they have been revealed to use an electron transport mechanism that creates positive redox potential by reversing electrons that were previously generated in the syntrophic reactions. In contrast, the identified affiliates of *Desulfarculales*, *Desulfobacteriales* and *Desulfovibrionales* phylotypes are sulfate reducers. However, in the absence of electron acceptors such as sulfate, the species are able to oxidize organic compounds, (e.g lactate, acetate and ethanol) into $CO_{2(g)}$ and $H_{2(g)}$ syntrophically [68]. The *Desulfuromonadales* species have been revealed to use $Fe^{3+}$ compounds to oxidize ethanol into $CO_{2(g)}$ and $H_{2(g)}$, but in strict co-operation with methanogens [56]. Lastly, the *Myxococcales* and *Chlamydiae* species have been identified as fermenters of carbohydrates [69] and proteins [70], which are ultimately converted into organic acids. All these *δ-Proteobacterial* roles contribute to the systems' stability by providing electrons, which act as reducing equivalents in the AD systems. We also postulate that differences in cow-dug's substrate composition were the main drivers of the observed variations in the *δ-Proteobacteria* abundances and β-diversities among the treatments (S6B Fig). The other low abundance species identified in our study were assigned to the *Mariprofundale*'s order, the affiliates of *Z-Proteobacteria* class. The affiliates co-exist with *Nitrosopumilales* to oxidize $Fe^0$ [71] into $Fe^{2+}$ or $Fe^{3+}$ ions. Moreover, the two phylotypes species have previously been revealed to convert nitrogen substrates into $NO_{3}^{-}{}_{(aq)}$ in their syntrophy [72,

73]. In this study, through their syntrophy, we hypothesize the completion of redox reactions in the treatments through electrons release. In summary, all affiliates of *Proteobacteria* identified in this study are capable of reversing oxidative damage to methionine [56], a trait that explains why their species are more (richness) compared to other phyla species in our treatments. Moreover, *Proteobacteria* species are able to express shorter diversification time-scales over other microbes, which lead to large population sizes, high growth rates, and strong selective regimes of the species. All these factors facilitate rapid *Proteobacteria* adaptation through either mutation or recombination events.

We also identified *Solibacterales* and *Acidobacterales* species that were affiliated to the phyla *Acidobacteria*. Members of these species ferment polysaccharides and monosaccharides into acetate and propionate [74]. The species are also able to fix $CO_{2(g)}$ anaplerotically [75], ultimately replenishing carbon atoms in the AD systems. Furthermore, the affiliates are also able to catabolize inorganic and organic nitrogen substrates as their N-sources [76]. The detection of these species in our study is attributed to the high amount of $NH_4^+{}_{(g)}$ in our AD systems. We postulate that reactor perturbations rather than environmental variables (Tables 2 and 3) were the main influencers of the genetic composition of the *Acidobacteria* in our study (S25B and S26B Figs).

Generally, the *Mollicute*'s species (*Tenericutes* phylum) detected in our study are facultative anaerobes. These were all absent in reactor 5, probably due to the nature of the reactor's substrates or feeds fed to the cattle breeds. The species are fermenters of sugars, and they convert the substrates into organic acids such as lactic acid, saturated fatty acids and acetate [77, 78]. All the identified species are affiliates of *Acholeplasmatales*, *Mycoplasmatales* and *Entomoplasmatales* which have previously been shown to exhibit reduced respiratory systems, with incomplete TCA cycle, that lack quinones and cytochromes. The metabolisms of *Tenericutes* generally yield low amount of ATP and large quantities of acidic metabolic end products [77]. It is probable that the produced acidic metabolites negatively affected the performance of other microbes in the systems in our study; which led to poor reactor performance. We hypothesize that the occurrence of these species in our systems was driven by the agroecological zones and soil types, which directly affected the nature of the cow-dug's substrates fed into our treatments (Tables 2 and 3 and S30 Fig).

Like other *Bacterial* species, the affiliates of *Actinobacteria* (S17A Fig) identified in this study, are able to metabolize a wide variety of substrates [79–81] to produce acetic acid, lactic acid, $H_{2(g)}$ and other products. Among the *Actinobacteria*, *Coriobacteriales* metabolize formate and $H_{2(g)}$, and concomitantly reduce $NO_3^-{}_{(aq)}$ into $NH_{3(g)}$ [79]. *Slackia Heliotrinireducens* was one of the *Coriobacteriales* species identified in this study. Species of the *Bifidobacteriale* were also identified in our study. These species are able to metabolize oligosaccharides and release lactic acid [80]. Other species belonging to *Actinomycetales* syntrophically metabolize propionate into acetate [81]. The metabolism of $NH_{4(aq)}$ by *Actinobacteria* is known to protect methanogenic *Archaea* from other oxidizing agents. However, their species have been reported to be more sensitive to nutrients perturbations [81] and due to this, reactor perturbations particularly the feeding regimes (S17B Fig) may have contributed to the observed species abundances and diversities among our treatments in this study. The low abundance of *Plactomycete*s in this study has also been observed elsewhere [82]. *Plactomycete*s have been found to metabolize sulphate-containing polysaccharides [82], recalcitrant hydrocarbons and other organic compounds [83]. However, when in syntrophic association with *Chloroflexi* (S18A–S21A Figs), the species are also able to oxidize butyrate [84] into smaller substrates.

The low abundance of communities of *Thermotogae* in this study is similar to the findings of previous studies [85]. The carbohydrate metabolism roles of the *Thermotogae* were previously determined based on the gene content of *Defluviitoga tunisiensis* [86]. The species have

been reported to convert the macromolecule into acetate, ethanol, $CO_{2(g)}$, and $H_{2(g)}$. In most cases, the species utilize the Embden-Meyerhof Parnas (EMP) pathway to generate pyruvate, which is either converted into acetyl-CoA, the precursor for acetate and ATP formation or reduced to ethanol, while at the same time oxidizing electron carriers. The $H_{2(g)}$ produced in these reactions is coupled to the oxidation of NADH and reduction of ferredoxin [86]. The reactions have been found to favour proton production which reduces the $P_{H2}$ in the AD systems. However, for these reactions to proceed favourably *Thermotogales* have been revealed to depend on *Clostridiales* species [87] which implies that the abundance and richness of *Thermotogales* is regulated by the *Clostridiales* species. The *Spirochaetes* were detected in low abundance in our study as was also found out by Wang and his co-wokers [88]. The identified *Spirochaetes* in this study are homoacetogens and acetogens which have previously been revealed to convert $H_{2(g)}$, $CO_{2(g)}$ and $CO_{(g)}$ into acetate [88, 89]. These species have also been revealed to co-exist with methanogens to provide an alternative metabolic route, termed as the butyrate oxidation pathway for $H_{2(g)}$ consumption [89]. Similarly, the *Spirochaetes* species have also been found to co-exist with the *Thermotogale's* species and to jointly ferment tetrasaccharides and cellobiose into organic acids [87, 90]. These syntrophies protect *Clostridiale's* species from substrate inhibition mechanisms [87]. Based on our findings and those reported elsewhere, we suggest that numerous microorganisms identified in our AD systems co-operate through the use of an assortment of mechanisms to metabolize the substrates. Due to this co-operation, we further postulate the existence of a holobiontic mechanism in our treatments, which could be regulating symbionts and other syntrophic species through ecological selection, via co-evolution to minimize conflict between or among the symbiotic species [91].

The *Verrucomicrobia* communities identified in this study were affiliated to *Methylacidiphiles*, *Puniceicoccales* and *Verrucomicrobiales* orders (S28A and S29A Figs). These communities have been established to metabolize cellulose and other $C_1$-containing compounds [92]. The affiliates of *Cyanobacteria* were classified as *Chroococcales*, *Oscillatoriales* and *Nostocales* species (S23A and S24A Figs) which are mainly involved in $H_{2(g)}$ and $N_{2(g)}$ metabolism [93–95] in addition to carbohydrate metabolism. Research has shown that the conversion of N-containing substrates into $NH_{3(g)}$ is an endergonic reaction that requires ATP [93–95]. The reaction has also been found to oxidize $H_{2(g)}$ into obligate $H^+_{(aq)}$ ions, circumventing high $P_{H2}$ in the AD systems [93–95]. Therefore, the recycling of $H_{2(g)}$ through the oxyhydrogen route, also termed as the Knallgas reaction by the *Cyanobacteria* species, has been revealed to provide ATP and protect strict anaerobes and other enzymes from oxidizing agents, by providing reducing equivalents and removing toxic $O_{2(g)}$ species [96, 97]. In our case, we attributed the variation in abundance of *Cyanobacteria* species among the treatments to the variability of $H_{2(g)}$ concentration in their respective treatments.

Another identified phylotype in our study was the *Candidatus poribacterial* species. These species ferment carbohydrate and proteins, including urea [98] into organic acids. The species have been revealed to use the Wood-Ljungdahl pathway also known as $CO_{2(g)}$ reductive pathway to fix $CO_{2(g)}$ generated from carbohydrates and other macromolecules. The species have also been previously reported to utilize the oxidative deamination pathway [99] to ferment amino acids, and the randomized methyl-malonyl-CoA pathway to metabolize propionate. All these reactions produce $H_{2(g)}$ and we postulate commensal symbiontism between *C. poribacteria* and acetogenic species in our treatments. Like other *Bacteria* species, *Fibrobacter* members such as *Fibrobacter succinogenes* identified in our study were likely to have been involved in the metabolism of cellulose [100] and other macromolecules in our AD systems.

Although it's not a common tradition to explain the existence of metal-reducing organisms in the AD systems, it is worth noting that the electron sink ability of any system depends on these organisms. One of such phylotypes identified in our study were the affiliates of

*Gloeobacterales* order (S22A and S23A Figs). The member species have previously been reported to transfer electrons directly from methanogens to syntrophic *Bacteria* [101]. These kinds of reactions are thought to enhance clean biogas production through the inhibition of sulfate reducing *Bacteria*. Other identified metal reducing species in our study were the members of *Clostridiales* that have previously been reported to utilize $Fe^{3+}$ $Co^{3+}$ and $Cr^{6+}$ as electron acceptor [101]. Other identified *Clostridiales*' species have also been reported to use the acrylate pathway to reduce arsenate and thiosulfate concomitantly fermenting organic molecules [102]. The affiliates of *Nitrospirales* (*Nitrospira* phylum) identified in this study, have been reported to reduce sulfate using $H_{2(g)}$ and thereby promoting acetate oxidation [103, 104]. We hypothesize that all these metal reducing reactions contributed to the reactor stability through the provision of reducing equivalents in the systems.

## The identified *Archaeal* biomes

The generated volatile organic acids, $CO_{2(g)}$ and $H_{2(g)}$ by acidogens, acetogens, and homoacetogens are further metabolized by *Archaea*, particularly the species that generate $CH_{4(g)}$ [105]. The *Archaea* domain constituted six (6) methanogenic and ten (10) non-methanogenic guilds identified in our study, with the two groups playing equally important roles in the AD systems. The methanogenic species are consumers of *Bacterial* products and convert them into $CH_{4(g)}$, the biogas. The methanogenic and non-methanogenic species identified in this study are affiliates of *Euryarchaeotes*, which utilize hydrogenotrophic, acetoclastic, methylotrophic and methyl-reducing pathways to metabolize the substrates of *Bacterial* species into $CH_{4(g)}$. Generally, the species of *Euryarchaeota* are known to indirectly stabilize the AD systems by consuming $H_{2(g)}$ [106] and we suggest that the varied $H_{2(g)}$ concentrations among the treatments, contributed to the observed variances in *Euryarchaeota*'s abundances and β-diversities among the local communities. However, environmental (Tables 2 and 3) and reactor perturbations may also have influenced the community abundances. Among the methanogens, *Methanosarcinales*, the affiliates of *Methanomicrobia* class, were the second most abundant group, consisting of 1–3% of the total communities identified. The affiliates of *Methanosarcinales* use three metabolic routes singly or in combination to convert methanol and methylamines, acetate or $CO_{2(g)}/H_{2(g)}$ and other potential substrates into $CH_{4(g)}$. The produced $H_{2(g)}$ from the reactions is also consumed by the same species to maintain $P_{H2}$ [107]. However, some species within the guild, are known to lower the accumulated $P_{H2}$ through the reassimilation of acetate, the process that transfers $H_{2(g)}$ to acetogens. Under certain conditions, the species of *Methanosarcinales* (e.g *Methanosarcina acetivorans C2A* and *Methanosarcina barkeri*) and *Methanobacteriales* (*Methanothermobacter thermoautotrophicus*) have been reported to express metabolic features similar to acetogenic organisms. Previously, these species have been reported to use acetyl-CoA pathway to metabolize $CO_{(g)}$ into acetate and formate, under different $P_{CO(g)}$ [108]. These reactions provides alternative route for ATP generation from $CO_{2(g)}$ reduction by the generated $H_{2(g)}$. Normally, the *Methanosarcinales* couple methanogenesis to ion transport, also termed as chemiosmotic coupling or chemiosmosis, to establish an electrochemical gradient across the cell membrane [108]; because the $CO_{2(g)}$ reduction reaction is normally unfavourable. The *Methanosarcinales* have however been reported to express significant metabolic plasticity.

Among the *Archaea* in our study, *Methanomicrobiale*s, affiliates of the *Methanomicrobia* class (S36A Fig) were the most abundant within the class, comprising of 2–4% of the identified total microbial populations. The affiliates of *Methanomicrobiale*s use two methanogenesis pathways; hydrogenotrophic and methylotrophic routes, to convert formate, $H_{2(g)}/CO_{2(g)}$, 2-propanol/$CO2_{(g)}$ and secondary alcohols [109] into $CH_{4(g)}$. The *Methanomicrobiales*

consume isopropanol to yield acetone and the reducing equivalent, NADPH [110]. However, the species are known to lack the $CO_{2(g)}$ reductive pathway, and hence uncouple methanogenesis from $CO_{2(g)}$ fixation, which implies that the species uses electron birfucation to conserve energy. Due to these characteristics, the *Methanomicrobiales'* species are able to express high acetate requirement phenotype for carbon assimilation [111]. In costrast, the identified *Methanocellale*'s species, which are affiliates of the same class, solely use the acetoclastic route to convert $CO_{2(g)}$ into $CH_{4(g)}$ [112]. Majority of the *Methanocellales'* species have been reported to reduce $CO_{2(g)}$ with $H_{2(g)}$, except in rare cases where formate and secondary alcohols are used as alternative electron donors. Similarly, all the identified *Methanobacteriales'* species, the affiliates of *Methanobacteria* class, use the $CO_{2(g)}$ reductive route to convert $CO_{2(g)}$ into $CH_{4(g)}$ with formate and $H_{2(g)}$ acting as electron donors. We are prompted to postulate that formate metabolism particularly in acetoclastic methanogenesis was triggered by the low $P_{H2}$ and that through electron bifurcation the species were able to conserve their energy. Other detected methanogenic species were the affiliates of *Methanopyrales*, belonging to *Methanopyri* class and *Methanococcales'* species that were assigned to *Methanococc*i class (S34A and S35A Figs). The members of the two classes have previously been reported to metabolize $H_{2(g)}$ and $CO_{2(g)}$ into $CH_{4(g)}$ under thermophilic and saline conditions [113, 114], which indicate that our treatments had high amounts of salts and that the temperatures were high enough to support the species.

In contrast, the non-methanogenic group of *Archaea* generally reduce metal elements and their compounds [101, 102], to provide reducing equivalents in the AD systems. The affiliates of *Halobacteriales* (*Halobacteria*, *Euryarchaeota)* were among the detected non-methanogenic species in this study. The member species of this group of non-methanogens use the arginine deiminase pathway [115, 116] or other metabolic routes to catabolize amino acid substrates. The *Halobacteriales* conserve energy when the intermediate metabolites; ornithine and carbamoyl-phosphate, are further converted into $CO_{2(g)}$ and $NH_{3(g)}$. The significant differences in relative abundances of the *Halobacteriales* species among the treatments can be attributed to the concentration of the alternative electron acceptors and salts concentrations in the systems. However, other deterministic factors such as competition and niche differentiation could also have led to the observed variations in *Halobacteriales* abundance among the treatments.

Our study also identified *Thermoplasmatales*, the affiliates of *Thermoplasmata* class. These groups of *Archaea* utilize sulfur for energy consevation [117]. However, their close relatives, *Methanomassiliicoccales*, formerly *Methanoplasmatales* have been found to use the methyl-reducing pathway to convert methanol into $CH_{4(g)}$. These relatives conserve energy through electron bifurcation [118] and we postulate that the affiliates of *Thermoplasmatales* are also involved in $CH_{4(g)}$ production, only that few or none of the biochemical assays have previously been conducted to test their biogas production abilities. We also identified *Thermococcales'* species, of the class *Thermococci*. The affiliates of this class of *Archaea* have been reported to have short generation time and to withstoond nutrient stress for longer durations [119]. Like for *Bacterial* species, and in the absence of sulfur, *Thermococcales*, have been reported to be involved in both acidogenesis and acetogenesis reactions. Species of this group metabolize proteins and carbohydrates into organic acids, $CO_{2(g)}$, and $H_{2(g)}$ [120, 121]. Other affiliates of *Thermococcales'* are involved in homoacetogenic reactions through which they oxidize $CO_{(g)}$ into $CO_{2(g)}$, and catabolize pyruvate into $H_{2(g)}$ and other products [122, 123]. We also identified *Archaeoglobales*, the affiliates of *Archaeoglobus* class which use sulfate, as the reducing agent, to oxidize lactate, long fatty acids and acetate into $CO_{2(g)}$ [124]. Some species e.g *Geoglobus ahangari*, however, reduce $Fe^{3+}$ compounds with $H_{2(g)}$ [125]. The identified *Archaeoglobales* and other metal reducing species identified in our study indicate the importance of these microbes in redox balancing reactions which contribute to reactor stability.

The *Sulfolobales* and *Desulfurococcales*, all belonging to *Thermoprotei* class and *Crenarchaeota* phylum were also identified in this study (S37A Fig). Species of these groups metabolize carbohydrates and amino acids [126, 127] into pyruvate and $CO_{2(g)}$. They also ferment pyruvate into lactate and $CO_{2(g)}$. The $CO_{2(g)}$ generated from the above reactions either enters into the $CO_{2(g)}$ reductive pathway, the route that is catalyzed by acetogens/homoacetogens to produce $CH_{4(g)}$ or is fixed through the 3-hydroxypropionate/4-hydroxybutyrate or dicarboxylate/4-hydroxybutyrate cycles by the identified *Thermoprotei* species to generate the energy [128]. Among other *Thermoprotei* affiliates identified in this study were *Thermoproteales* that use sulfur and its compounds as electron acceptors, to ferment carbohydrates, organic acids and alcohols [129]. In these reactions, the species consume $CO_{2(g)}$ and $H_{2(g)}$. Though the species have been previously reported to grow at low rate, they also ferment and oxidize amino acids and propionate, respectively [130]. Other affiliates like *T. neutrophilus* are known to assimilate acetate and succinate in the presence of $CO_{2(g)}$ and $H_{2(g)}$ [120, 131]. This species use $H_{2(g)}$ to reduce sulfur. Unlike most of the other *Archaeal* species, *Thermoproteales* use oxidative and reductive TCA routes to conserve energy. *Acidilobales* another *Thermoprotei* order identified in this study uses $S^0$ and thiosulfate to metabolize protein and carbohydrate substrates [132]. The species use either EMP or the Entner-Doudoroff (E-D) [133] pathways to convert the substrates into acetate and $CO_{2(g)}$ [134]. The affiliates of *Acidilobales* are also able to oxidize triacylglycerides and long-chain fatty acids [134] into acetate and $CO_{2(g)}$. Substrate-level and oxidative phosphorylation are the major routes of energy conservation in *Acidilobales*; and the two mechanisms occur when $S^0$ and acetyl-CoA concentrations are high. Due to $S^0$ respiration in *Acidilobales* species, we postulate that their member species close the anaerobic carbon cycle via complete mineralization of organic substrates and the variations of these substrates together with other deterministic and non-deterministic (Tables 2 and 3) factors contributed to the observed variation in abundance among the treatments and the high β-diversity (S37B Fig). We also hypothesize that soil type was another factor that exerted natural selection on the *Acidilobales* through the alteration of the cow-dung's substrates chemical composition.

The identified *Thaurmarchaeota* communities were classified into *Cenarchaeles* [135] and *Nitrosopumilales* [136] phylotypes (S38 Fig). The member species of these groups use the modified 3-hydroxypropionate/4-hydroxybutyrate pathway to fix inorganic carbon to generate energy. The communities are also able to metabolize urea and its products into, $NH_{3(g)}$ [137, 138]. The *Nanoarchaeota* (S31A Fig) species exploit *Crenarchaeota* mechanisms [139, 140] to metabolize amino acids and complex sugars [141, 142] while the *Korarchaeotes* are sole amino acid fermenters. The energy metabolism in the last two phyla is still unclear and further biochemical studies are needed to unravel their full metabolic activities.

## The identified *Fungal* biomes

Like the *Bacterial* species described in this study, the identified *Fungi* (S39A–S41A Figs) have the ability to convert macromolecular substrates into acetate, $CO_{2(g)}$ and $H_{2(g)}$ [143]. Among the identified *Fungal* species, *Ascomycota* were the richest and most abundant guild, comprising of <1% of the identified communities in the treatments. Among the *Ascomycota* species, a majority were affiliates of *Saccharomycetales* (*Saccharomycetes*, *Ascomycota*), that generally use the EMP pathway to ferment glucose and other sugars into pyruvate. One of such identified species is *Saccharomyces cerevisiae* [144] that is known to ferment carbohydrates into mixed fermentation products. Other *Saccharomycetes* species e.g *Hansenula polymorpha* yeast ferment monosaccharides, fatty acids and amino acids into acetate, ethanol, $CO_{2(g)}$ and $H_{2(g)}$ [145]. Furthermore, the species are also able to oxidize methanol into formaldehyde and $H_2O_{2(g)}$ [146]. The two products act as the sources for formate and $H_{2(g)}$, the precursors for

methanogenesis. The observed variations in abundance of the *Saccharomycetes* species may be attributed to their metabolic plasticity, deterministic factors and other environmental variables. The other *Ascomycota* affiliates belonging to *Pezazales* (*Pezizomycetes*), *Pleosporales* (*Dothideomycetes*), and *Eurotilaes* (*Eurotiomycetes*) identified in this study use mixed fermentation pathways to ferment carbohydrates [147, 148]. The metabolisms of these *Ascomycota* species have however been observed to vary within and among the phylotypes, despite the presence of a common enzyme for the pyruvate metabolism [147, 148]. For example, among the *Eurotiale*'s order; the species *Paecilomyces variotii* expresses low metabolic capacity on plant-derived compounds compared to other *Eurotiale*'s members [149, 150]. This seems to indicate the genetic diversity of *Eurotiomycetes*' species, seen in the variations of abundance of the species among the treatments in this study. Our data also suggests that the communities of *Eurotiomycetes* were influenced by the environmental temperature and soil type (Tables 2 and 3 and S45A Fig). Like the identified *Fusobacteria* species and other identified amino acid metabolizers, *Dothideomycetes*, another amino acid consuming phylotype, metabolize lipids and proteins into volatile organic acids [150]. On the other hand, the affiliates of *Pezizomycetes* use sulfate as an electron sink to consume amino acids molecules [151, 152]. We attribute the community variation observed within the above groups to the composition of the cow-dung's substrates and natural selection pressure differences among the treatments.

The *Sordariomycetes*' affiliates identified in this study (S44A Fig) produce enzymes that express an assortment of degradative mechanisms and other fermetative roles. The *Hypocreales*' affiliates are such an example that produces powerful hydrolytic enzymes [153] that catabolize complex biomass in their vicinity. *Fusarium* sp. a member of *Hypocreales*, metabolizes $NH_{3(g)}$, lipids and proteins [154, 155] into their respective smaller molecules. The species has been found to use NAD(P)H to reduce $NO_3^-{}_{(aq)}$ into $NH_4^+{}_{(aq)}$. The production of $NH_4^+{}_{(aq)}$ by *Hypocreales* is normally coupled to the substrate-level phosphorylation and ethanol oxidation; the processes that produce acetate [154, 155]. The two mechanisms ameliorate the $NH_4^+{}_{(aq)}$ toxicity to micro-organisms. Other *Hypocreales* species such as *Hirsutella thompsonii* confer wide metabolic activities that involve chitinases, lipases, proteases and carbohydrate degrading enzymes [156]. *Monilia sp.* [157] and *Neurospora crassa* [158] were among the identified *Sordariomycetes* and they convert cellulose into ethanol. *Magnaporthales* such as *Pyricularia penniseti* convert isoamylalcohol into isovaleraldehyde while other *Magnaporthales*' species convert fumaryl-acetoacetate into fumarate and acetoacetate [159, 160]. Other roles played by the *Sordariomycete*s' species include the metabolism of aromatic compounds through which, phenolics and aryl alcohols/aldehydes are converted into acetate, formate and ethanol [161, 162]. Our data suggests that *Sordariomycete*s community's abundances were influenced by the cow-dug's substrates and the environmental variables (Tables 2 and 3) of the treatments.

In contrast to the *Ascomycota* species, affiliates of *Basidiomycota* establish susceptible hyphal networks [163] that may have been constrained by the reactor pertubartions in our study. This may explain why the *Ascomycota* species dominated over the *Basidiomycota*'s species in this study. Nonetheless, like *Ascomycota* species, the affiliates of *Malasseziales*, *Ustilaginales*, *Agaricules* and *Tremellales* species, all belonging to *Basidiomycota*, are able to metabolize carbohydrates [148, 164], urea [164] and $NO_3^-{}_{(aq)}$ [164] in the systems. Other members like *Polyporales* metabolize woody substrates and their respective products [165]. All these functions clearly indicate that *Basidiomycota's* species played diverse metabolic roles in our AD systems in this study. Interestingly, we also identified *Chytridiomycota* affiliates in reactor 9 and 10. These affiliates are capable of catabolizing pollen, chitin, keratin, and cellulose [166, 167], which may be indicative of the presence of these substrates in the two treatments. The detected *Spizellomycetales*, *Cladochytriale*s and *Rhizophydiale*s species in the two

treatments, further revealed the presence of highly decayed plant tissue with high cellulose content. The identified *Monoblepharidomycetes* members are close relatives of *Cladochytriales* and *Spizellomycetales* [168] species and were detected only in reactor 9 and we postulate that these groups perform similar roles in the AD systems.

Generally, *Chytridiomycota* species are carbohydrate metabolizers as previously revealed by the presence of glucose transporters and invertase activity in their systems [166, 167]. The species use electron bifurcation mechanisms to conserve the energy and the mechanisms occur in the hydrogenosome [169]. The *Blastocladiales*, the close relatives of *Chytridiomycotes* and the affiliates of *Blastocladomycota*, class *Blastocladomycetes* were only detected in reactor 9 in our study. Their member species metabolize cellulose into acetate, $CO_{2(g)}$ and $H_{2(g)}$. Interestingly, *Blastocladiales* have previously been reported to use the Malate fermentation pathway to generate and conserve energy [170]. *Mortierellales* formerly classified under *Mucoromycotina* [171] and latter assigned to *Mortierellomycotina* phylum [172] are converters of organic compounds into poly-unsaturated fatty acids [173] while their relatives *Entomophthorales* and *Mucorales* metabolize decaying organic matter. However, the affiliates of the three orders were only detected in reactor 9, 10 and 11, respectively.

Like similar studies [174, 175], our data revealed that *Fungi* co-occurred with the species of *Bacteria* and methanogens. In these syntrophy, we postulate that the *Fungal* species catalyzed inter-species $H_{2(g)}$ transfer and re-generated oxidized reducing equivalents; $NAD^+$ and $NADP^+$ in the systems. All these reactions increase metabolism of the dry matter [176] in the treatments. We also suggest that the *Fungal* syntrophy is more complex than simple cross-feeding mechanisms, because the species shift more oxidized products into a more reduced form. Through these mechanisms, the precursors, lactate and ethanol are converted into acetate and formate depending on the prevailing conditions. The involved interactions between *Fungi* and other domains' species have been determined to be very crucial to an extent of influencing the isolation and culturing of some *Fungal* species [176, 177]. The findings indicate why *Fungal* species should be jointly considered with other domains' species in the context of biogas production.

## The microbiomes diversity and biogas production

To establish whether environmental conditions and perturbations influenced the microbial populations, we compared the collected phenotypic traits with the genotypic dataset. The phenotypic datasets revealed an insight on the cause of the observed communities' β-diversities and biogas production variations. We attempted to use the principles of the four ecological mechanisms that include selection, drift, speciation, and dispersal to better understand the cause of the observed variance in microbial composition among the treatments and thus the cause of the biogas production variability among the sites. While selection is deterministic and ecological drift is stochastic, dispersal and speciation may have contributed to the two ecological parameters. One of the challenges of examining communities' in any given niche is the difficulty of estimating real dispersal among the sites. Nonetheless, our treatments were fluidic, naturally encouraging rapid microbial dispersion. Hence, species dispersal was not the major limiting factor that influenced our local communities. Similarly, we also excluded speciation based on our sampling process and experimental design. Therefore, we suggest that selection and drift were the likely two main factors that contributed to variation of community abundances among the treatments and the observed high β-diversities in our study. The two factors were most likely influenced by the substrate inputs (Table 3) and disturbances from environmental variables (Tables 2 and 3). Environmental conditions directly affect the chemical composition of the substrate fed to the cattle breeds. As a result, their wastes, the cowdung is fed to

the reactor which ultimately influences the microbial composition and biogas productivity. Temperature, agroecological zones and soil types were among the environmental factors that were observed to drive the specific communities' β-diversities and their relative abundances among the treatments. All these could have contributed to the observed variation of biogas production among the studied treatments. We postulate that environmental temperatures combined with other deterministic factors such as the species metabolic traits, largely contributed to the lack of identification of the *Chytridiomycota*, *Blastocladiomycota* and other rare *Fungal* species in the majority of the treatments. The findings increased the number of the total species in the few treatments, where the rare species were identified as opposed to where they were absent. Nonetheless, we never observed significant differences in reactor performances between the two categories of the treatments. Consequently, we linked the observations with the low abundances of the rare *Fungal* species identified in the few respective reactors, which implies the importance of the species abundance in biogas production. The composition of the cow-dung's substrates fed to these treatments could largely be influenced by cattle breeds and several environmental factors including agroecological zones, soil type, rainfall, environmental temperatures to mention but a few, and its chemical compositions could have stimulated the growth of specific species, favouring colonization and speciation in the respective treatments. Similarly the same phenomenon could have happened to other microbial communities. In this case, the available resources could have reduced competition among the species, weakening the niche selection abilities. Generally, these ecological factors normally strengthen the ecological drift and prority effects [178] that result to unpredictable site to site variation of microbial abundances and high β-diversities, even under similar environmental conditions. We used the same arguments to explain our observations on the variations of the relative abundances among treatments and high β-diversities, particularly when the conditions were similar (Tables 2 and 3). We also extended the same argument to the biogas production capacities among our treatments.

Besides the cow-dung's substrate effects, we hypothesize that the intensity and duration of the disturbances [179], led to high rate of species mortality, low growth rates and species extinction. These kinds of disturbances caused decrease of stochasticity and to account for their effects on our findings, we adopted the concepts of neutral theory [179]. Generally their associated caveats reduce ecological drifts that result to low competition abilities among the low abundance species. Due to these facts, we postulate that the persistence and intensities of these disturbances in our specific treatments eliminated weak and less adapted microbial populations. One of such identified case was revealed in reactor 5 in which the *Tenericutes* affiliates were absolutely extinct. Similarly, the absence of *Mortierellales*, *Entomophthorales*, *Mucorales* to mention but a few, in other treatments (S51 Fig) was an indication of unfavorable disturbances that could have originated from either cow-dung's substrates or other reactor perturbations. All these may have led to more predictable site-to-site variations and we used these concepts to explain the observed low β-diversity among the treatments (e.g in reactor 2 and 10, reactor 3 and 6 etc). However, to understand better the links between the environmental conditions and the microbial communities in these AD systems, we suggest the utilization of the metacommunity theory that considers the world as a collection of patches that are connected to form a metacommunity through organisms' dispersal. The model depends on the specific species traits, the variance of the patches in the environmental conditions as well as frequency and extent of species dispersal in the systems. Other ecological models lack these features. Therefore, the model can be used to ask and test whether dispersal, diversification, environmental selection and ecological drift were the vital elements in the biogas production process. In this case, a detailed understanding of the contribution of these factors on specific

species abundances and community assemblage can be used to tailor the reactor management from an ecological point of view.

## Conclusions and future prospects

Our results provide an insight into the biogas microbiomes composition, richness, abundances and variations within and among the taxa and the twelve reactor treatments. Interestingly, microbiomes structure and richness were similar among the twelve treatments, except for the affiliates of *Rhizophydiales*, *Cladochytriales*, *Monoblepharidales*, *Blastocladiales* and *Entomophthorales*, which were detected only in reactor 9. Contrasting the findings were the affiliates of *Mortierellales* and *Mucorales* that were identified in reactor 10 and reactor 11, respectively, while the *Spizellomycetales* species were detected in both reactor 9 and 10. We also revealed that the affiliates of *Tenericutes*, were the only *Bacterial* guild that was absent in reactor 5. The Cow-dung's substrates and metabolic capabilities of these species were the main elements that were considered to limit their dispersal to other treatments. Nonetheless, the identified phylotypes richness within and among the phyla were found to vary, except for the phyla, that comprised a single phylotype. All the identified species abundances among the treatments and the community's β-diversities in the majority of the treatments significantly varied. All these contributed to our conclusion that environmental and spatial variables largely influenced biogas microbiome populations and thus the amount of biogas produced. The discoveries highlight the importance of identifying the involved natural processes that give rise to such identified variations. Based on the results obtained in this study, we suggest further studies on species responses on the natural conditions for optimal biogas production and microbial community managements. We also suggest the application of metacommunity theory among other ecological theories in developing management strategies for the AD systems.

## Materials and methods

### Ethics considerations

Before embarking on the research, the study protocols were approved by the Institutional Research and Ethics committee (Ref. No. IREC/2017/52) of Moi University. Permission for conducting research was granted under the permit No. NACOSTI/P/17/86022/18890 by the National Commission for Science, Technology and Innovation, Kenya. The procedures were also reviewed and approved through the University of Eldoret [Ref. No. UoE/B/DVASA/REG/97] Research and Innovation Committee and the Institute for Biotechnology Research Board, JKUAT [Ref. No. JKU/BR411-1663/2014]. The consents to sample the biogas reactor treatments (installed in privately owned premises, Table 2), were sought orally from the reactors' owners. The collected phenotypic informations for the studied reactor treatments were specifically used for the purpose of the research and the owners are updated on the research findings. All other experiments were conducted in secured environments and in compliance with the biosafety and biosecurity requirements.

### Experimental design

Randomized complete block design was utilized as previously described [180], to assign the twelve cowdung digesting treatments to the four blocks, which operated at mesophilic conditions. Organic loading rate and the hydraulic retention time were largely influenced by the treatment size and the farmers' routine activities. The sludge sampling was conducted following the previously described procedure of Boaro *et al.* [181]. The treatments' gas volumes ($M^3$) were recorded using their installed gas metres at the farmers' premises. The treatments'

performances ($M^3$) were simulated through computing and integrating the reactor volumes ($M^3$) and the farmers actual collected gas ($M^3$). Geographic co-ordinates (Table 2) and agro-ecological zones (Table 3) were generated by Global Positioning System (etrex, Summit$^{HC}$, Garmin) and ArcGIS® software (ver10.1) [182], respectively. A pre-tested questionnaire was utilized according to Oksenberg et al. [183] protocol to collect the phenotypic dataset. Small scale farming and zero grazing of Freshian breeds were the main activities among the sampled farmers. However, the owners of reactor 3 and 11 (Meru and Kiambu blocks, respectively), reared Jersey breeds of cattle. Samplings were conducted during the warm and cool climatic conditions.

## Sample preparation and sequencing

The double stranded deoxyribonucleic acids (dsDNA) was extracted from the duplicate sludge samples using ZR soil microbe DNA kit (Zymo Research, CA., USA), according to manufacturers instructions. The quantity and quality of the extracted dsDNA were determined as previously described by Campanaro et al. [184]. The dsDNA was normalized and library preparation performed using Nextera$^{TM}$ DNA sample prep kit (Illumina, San Diego, CA, USA), according to manufacturers instructions. Library normalization was conducted using Illumina TruSeq DNA PCR-free sample preparation kit buffer (Illumina, USA). Quantification for the libraries was conducted by Qubit$^{TM}$ 2.0 (Invitrogen, Life Technologies, Carlsbad, CA., USA) and Bioanalyzer 2200 (Agilent technologies, USA). To prepare for cluster generation and sequencing, equal volumes (4nM with EBT buffer, 10mM TRIS, 0.1%Tween20) of normalized libraries were pooled, diluted in hybridization buffer (HT1) and denatured according to Nextera$^{XT}$ protocol. PhiX control spikes were also added. De novo deep sequencing was conducted by an Illumina MiSeq system as previously described by Caporaso et al. [185] but adopting a v3 kit chemistry with 300 cycles (2×150) as previously described [184].

## Bioinformatics and statistical analysis

All paired-end reads in FASTQ format were pre-processed with FastQC software (ver0.11.5) following the procedure described by Zhou et al. [186]. Trimmomatic software was utilized to trim and filter the adaptor according to Bolger et al. [187]. De novo assembly of the paired-end contigs were peformed by MetaSpade software (ver3.10) as described previously [188]. To re-trim low-quality regions of the scaffolds, a bit-masked k-difference matching algorithm, termed as Skewer software [189], was utilized. SolexaQA was also used to remove the regions, as described by Cox et al. [190] while DRISEE, metagenomic quality assessment software was further used to remove duplicated scaffolds according to Keegan et al. [191]. Denoising and normalization of the scaffolds, was conducted using the fastq-mcf [192]. We screened near-exact scaffolds that matched human, mouse and cow genomes following Langmead et al. procedure that used a sort read aligner, the Bowtie2 [193]. M5nr [194], a non-redundant tool, FragGeneScan [195] and BLAT [196] tools were utilized to compute datasets against the sixteen (16) publicly available biological databases; RefSeq [197], IMG [198], TrEMBL [199], Subsystems [24], KEGG [200], GenBank [201] SwissProt [202] PATRIC [203], eggNOG [204], KO [205], GO [206], COG [207], RDP [208], LSU [209] SSU [210] and NOG [211], as previously described by Wilke et al. [194]. 70% identity cut off value, was set to filter the rRNA genes from a reduced version of M5RNA [194] using SortMeRNA software [212], while the CD-HIT algorithm [213] was used to cluster the identified RNA and the predicted amino acid sequences at 97% and 90% identity, respectively. Subsequently, we assigned our sequences to the archived taxonomic profiles at nucleotide, protein and process levels of annotations. Two ENVO ontologies: large lake and terrestrial biomes were utilized according to Field et al. [214]

and Glass *et al.* [215]. Alignment similarity cut off values, were set at single read abundance; $\geq$60% identity, 15 amino acid residues and maximum e-value of $1e^{-5}$. The annotated datasets were archived in CloVR [216]. Prior to statistical analysis, the rarefaction curves were reconstructed, datasets was $\log_2$ (x+1) transformed and normalized using an R package, DESeq software [217]. PCoA based on euclidean model [218] was set at $P\leq0.05$, and conducted against the twelve treatment samples. Results were visualized through MG-RAST pipeline (ver4.01) [219] and Krona radial space filling display [220]. Optimal scripts and/or command lines and restful application programming interface were utilized in our analysis.

## Supporting information

**S1 Table. The quality control statistics of the obtained scaffolds for the twelve treatments.** (PDF)

**S1 Fig.** Barchart showing unfiltered and filtered sequencing reads (a) and the known and unknown protein genes (b) in our samples. More than 53.07% of the filtered nucleotide reads in our samples contained unknown proteins. (PDF)

**S2 Fig.** Stacked barchat showing some (21 phyla) of the identified *Bacteria* domain phyla, relative abundances (a) and the PCoA plot (phylum level), based on the Euclidean model (b). The communities' nucleotide composition of reactor 2 and 10 and those of reactors 4 and 9 partially clustered in the upper left and right quadrants of the plot respectively. (PDF)

**S3 Fig.** The PCoA plot showing the relative abundances variation at the class (a) and order level (b). At the class and order level, the nucleotide composition of reactor 2 and 10, were positioned on the lower left quadrant of the plot, those of reactor 3 and 6, on the upper left quadrant, the communities of reactor 8 and 11, on the upper right quadrant, while the composition of reactor 4 and 9, were postioned on the lower right quadrant of the plot. All clustered partially at both level. (PDF)

**S4 Fig.** The stacked barchart revealing six classes of *Proteobacteria* communities, relative abundances (a) and their PCoA plots based on the Euclidean model (b). The nucleotide composition for reactor 1, 3 and 6 clustered on the upper right quadrant of the plot. Similarly, the composition of reactor 4, 7 and 12 clustered partially on the lower right, while those of reactor 5 were singly positioned on the upper left quadrant of the plot. (PDF)

**S5 Fig. The PCoA plot based on the Euclidean model for all the identified *Proteobacteria* orders.** The PCoA plot indicated partial similarities of the nucleotide composition in half of the studied treatments, including reactor 1, 3 and 6, upper right quadrant; reactor 4, 7 and 12, lower right quadrant of the plot at the class level. Similar observations were made at the order level, with an exception of few treatments (reactor 4, 7, 8 and 9 that formed cluster). (PDF)

**S6 Fig.** Stacked barchat showing seven *δ-Proteobacteria* orders, relative abundances (a) and their PCoA plot revealing nucleotide composition variations, based on the Euclidean model (b). The plot revealed dissimilarity of nucleotide composition among the treatments, with an exception of the composition of reactor 4 and 7 that were found to cluster on the lower left

quadrant of the plot.
(PDF)

**S7 Fig.** Stacked barchart showing fourteen *γ-Proteobacteria* orders (a) and their PCoA plot based on the Eucleaden model (b). The plot revealed clustering of the reactor 1 and 3 nucleotide composition, which were in close proximity with the composition of reactor 7. The communities of reactor 7 clustered partially with those of reactor 8 and 12. However, the composition of reactor 4 and 9 were in close proximity. Interestingly, all the clustered treatments were positioned on the upper right quadrant of the plot. Moreover, the nucleotide compositions of reactor 5 were singly positioned on the upper left quadrant of the plot.
(PDF)

**S8 Fig.** Stacked barchat showing seven *β-proteobacteria* orders, relative abundances (a) and their PCoA plot revealing nucleotide composition (dis)similarities among the treatments (b). The plots revealed partial clustering of reactor 7 and 8 nucleotide compositions while the nucleotide composition of reactor 3, 6 and 12 were found to cluster on the lower left quadrant of the plot. The majority of the treatments comprised dissimilar *β-Proteobacteria* nucleotide compositions.
(PDF)

**S9 Fig.** Stacked barchat showing the seven *α-Proteobacteria* orders, relative abundances (a) and their PCoA plots, revealing the nucleotide composition variation among the reactors based on the Euclidean model (b). The plots revealed close proximity of reactor 1 and 6 nucleotide composition and were positioned on the y-axis. However, the composition of reactor 3 and 11 clustered partially on the upper right quadrant while the nucleotide compositions of the other treatments were found to reveal dissimilarity.
(PDF)

**S10 Fig.** Stacked barchat showing two *ε-Proteobacteria* orders, relative abundances (a) and their PCoA plot based on Euclidean model (b). The PCoA plot for reactor 4 and 8 clustered on the upper left quadrant of the plot while those of reactor 5 and 9 clustered partially on the upper right quadrant.
(PDF)

**S11 Fig.** The stacked barchat revealing four *Firmicutes* classes, the relative abundances (a) and their PCoA plots, revealing variance among the twelve reactors based on the Euclidean model (b). The PCoA plots revealed close proximity of reactor 4 and 9 and reactor 1 and 8 nucleotide compositions, while the reactor 1 and 7 composition clustered partially on the plot. However, those of reactor 2, 10 and 12 were found to cluster on the lower left quadrant of the plot while other treatments' compositions were found to reveal dissimilarity, except the composition of reactor 5 that were singly positioned on the upper left quadrant of the plot.
(PDF)

**S12 Fig.** Stacked barchat showing eight *Firmicute's* orders, relative abundances (a) and their PCoA plot based on the Euclidean model (b). The plot revealed partial similarities between the composition of reactor 2 and 10, reactor 3 and 6, and reactor 7 and 12 while the nucleotide composition of reactor 1 and 9 almost clustered.
(PDF)

**S13 Fig.** Stacked barchat showing the four *Clostridia* orders (a) and their PCoA plots, revealing nucleotide composition variations among the twelve reactors based on the Euclidean model (b). The plots revealed dissimilarities of the nucleotide composition among the majority

of the treatments. However, the composition of reactor 5 and 9 were clustered partially on the upper left quadrant, while the nucleotide of reactor 7, 11 and12 clustered on the lower left quadrant of the plot.
(PDF)

**S14 Fig.** Stacked barchat showing two *Bacilli* orders, relative abundances (a) and their PCoA plot based on the Euclidean model (b). The nucleotide composition varied among the treatments and pairwise partial clustering of the nucleotides was observed among the majority of the treatments, with an exception of reactor 5 and 8 communities that were distinctively positioned within the plot.
(PDF)

**S15 Fig.** The stacked barchat revealing four *Bacteroidete*s classes, relative abundances (a) and their PCoA plots revealing nucleotide composition variation, based on the Euclidean model (b). The plots revealed partial clustering of reactor 2 and 10 and reactor 8 and 12 communities on the plot. However, the communities of reactor 4 and 9 were found to cluster on the lower right quadrant of the plot.
(PDF)

**S16 Fig.** Stacked barchat showing the four *Bacteroidete*'s orders, the relative abundances (a) and their PCoA plot based on the Euclidean model (b). The nucleotide composition of reactor 2 and 10 and those of reactor 8 and 12 clustered partially on the lower left quadrant of the plot while the composition of reactor 4 and 9 were revealed to cluster on the lower right quadrant of the plot.
(PDF)

**S17 Fig.** Stacked barchat showing six *Actinobacteria* orders, the relative abundance (a) and their PCoA plot based on the Euclidean model (b). The nucleotide composition of reactor 7 and 8 positioned on the upper right quadrant of the plot were found to cluster partiall, while the nucleotide of reactor 1, 3, 5, 11 and 12 formed a cluster along the y-axis of the plot.
(PDF)

**S18 Fig.** The stacked barchat showing four *Chloroflexi* classes, relative abundances (a) and PCoA plot revealing variance among the twelve reactors, based on the Euclidean model (b). The plots revealed close proximity of reactor 1 and 7 nucleotide composition, and were positioned on the upper left quadrant of the plot. However, the composition of reactor 4, 5, 10 and 12 were found to cluster on the lower right quadrant of the plot.
(PDF)

**S19 Fig.** Stacked barchat showing five *Chloroflexi* orders, relative abundances (a) and their PCoA plot based on the Euclidean model (b). The nucleotide composition of reactor 4, 5, 10 and 12 clustered, on the lower right quadrant of the plot, while the composition of reactor 1 and 7 were found to reveal close proximity on the upper left quadrant of the plot.
(PDF)

**S20 Fig.** Stacked barchat showing two *Chloroflexi* class orders, relative abundances (a) and their PCoA plot based on their Euclidean model (b). The nucleotide composition of reactor 1 and 5 (clustered; upper right quadrant) and those of reactor 3 and 7 (clustered; lower right quadrant) were found to similar. Notably, the nucleotide reads of reactor 1 and 10 were revealed to cluster partially with the composition of reactor 3 and 7. Similarly the nucleotides of reactor 8, 11 and 12 clustered partially on the upper left quadrant of the plot.
(PDF)

**S21 Fig.** Stacked barchat showing the two *Thermomicrobia* orders, relative abundances (a) and their PCoA plot based on the Euclidean model (b). The PCoA plot revealed dissimilarities of the nucleotide compositions among the twelve studied treatments.
(PDF)

**S22 Fig.** Stacked barchat showing *Cyanobacteria* class, relative abundances (a) and their PCoA plot based on the Euclidean model (b). The plot revealed dissimilarities of the nucleotide composition among the treatments and only the nucleotides of three treatments (reactor 3, 7 and 10) that were found to be distinct within the plot.
(PDF)

**S23 Fig.** Stacked barchat showing the five *Cyanobacteria* orders, relative abundances (a) and their PCoA plot based on the Euclidean model (b). The PCoA plot revealed four out of twelve reactors that were distinctively located in plots. Other treatments formed partial clusters or clustered within the plots.
(PDF)

**S24 Fig.** Stacked barchat showing the four affiliates of the Unclassified *Cyanobacteria* nucleotide reads, relative abundances (a) and their PCoA plots based on the Euclidean model (b). The plot revealed dissimilarities among the treatments, all distributed in the four plot quadrant.
(PDF)

**S25 Fig.** The stacked barchat showing the two *Acidobacteria* classes, relative abundances (a) and their PCoA plots for their nucleotide composition based on the Euclidean model (b). The nucleotide composition of reactor 1 and 3 clustered partially on the lower right quadrant of the plot, reactor 10 and 11 compositions were positioned in close proximity on the upper right quadrant of the plot while the nucleotides of reactor 2, 8 and 12 were found to cluster on the upper right quadrant of the plot.
(PDF)

**S26 Fig.** The stacked barchat showing the two *Acidobacteria* orders, relative abundances (a) and their PCoA plots for their nucleotide composition based on Euclidean model (b). The composition of reactor 1 and 3 clustered partially on the lower right quadrant of the plot, those identified in reactor 10 and 11 were in close proximity, positioned on the upper right quadrant of the plot while the composition of reactor 2, 8 and 12 clustered in the upper right quadrant of the plot.
(PDF)

**S27 Fig.** Stacked barchat showing two *Deinococcus-Thermus* orders, the relative abundances (a) and the PCoA plots based on the Euclidean model (b). The PCoA plots revealed clustering of the nucleotide composition of reactor 3 and 6, and those in reactor 7 and 9 on the upper left quadrant and lower left quadrant of the plot respectively. However, those identified in reactor 4 were in close proximity with those of reactor 7. The composition of reactor 11 and 12 partially clustered on the lower right quadrant of the plot. The nucleotides of reactor 2 and 10 were the only nucleotide reads located in the upper right quadrant of the plot.
(PDF)

**S28 Fig.** Stacked barchat showing *Verrucomicrobia* classes, relative abundances (a) and their PCoA plot based on the Euclidean model (b). The PCoA plot revealed dissimilarities of the nucleotide composition among the treatments, except those identified in reactor 3 and 7 that

clustered partially on the lower right quadrant of the plot.
(PDF)

**S29 Fig.** Stacked barchat showing three *Verrucomicrobia* orders, relative abundances (a) and their PCoA plot based Euclidean model (b). The composition of reactor 3, 6, 7 and 11 clustered partially, on the lower left quadrant of the plot; those indentified in reactor 1 and 10 were also clustered partially, on the upper right quadrant, while the reads of reactor 5 and 8 were found to cluster on the upper right quadrant of the plot.
(PDF)

**S30 Fig.** Stacked barchat showing three *Ternericutes* orders, relative abundances (a) and their PCoA plot based on the Euclidean model (b). The nucleotide composition in reactor 4 and 9 and those detected in reactor 6 and 8 partially clustered in the upper right quadrant of the plot. Other treatments comprised dissimilar nucleotide composition.
(PDF)

**S31 Fig.** The stacked barchart showing the four *Archaea* phyla, relative abundances (a) and their PCoA plot based on the Euclidean model (b). The compositions of reactor 4 and 12 clustered, on the lower left quadrant of the plot; while those of reactor 3 and 6 and reactor 7 and 9 clustered partially; on the lower right quadrant and upper left quadrant of the plot. Other treatments were distinctively dissimilar, positioned within the plot.
(PDF)

**S32 Fig.** Stacked barchat showing nine *Archaea* classes, and the affiliates of the unclassified reads, relative abundances (a) and their PCoA plot based on the Euclidean model (b). The nucleotide compositions in the respective treatments were distinctively dissimilar.
(PDF)

**S33 Fig.** Stacked barchat showing sixteen *Archaea* orders, relative abundances (a) and their PCoA plot based on the Euclidean model (b). The nucleotide of reactor 3 and 7, were clustered partially on the lower left quadrant of the plot. However, the compositions of the other ten treatments were distinctively dissimilar, all distributed in the plot.
(PDF)

**S34 Fig.** The stacked barchat showing the eight *Eurychaeota* classes, the relative abundances (a) and their PCoA plots revealing dissimilarities of the nucleotide composition based on the Euclidean model (b). The nucleotide composition of reactor 1 and 7, were positioned on the upper left quadrant of the plot while the composition of reactor 6 and 8, were positioned on the lower left quadrant, near the y-axis were in close proximity.
(PDF)

**S35 Fig.** Stacked barchat showing ten *Euryarchaeota* orders, relative abundances (a) and their PCoA plot based Euclidean model (b). Their nucleotide compositions were found toreveal dissimilarity, all distributed within the four quadrants.
(PDF)

**S36 Fig.** The stacked barchat showing three *Methanomicrobia* orders, the relative abundances (a) and their PCoA plot based on the Euclidean model (b). The PCoA plot revealed partial clustering of reactor 1 and 7 nucleotide composition on the upper left quadrant of the plot. Further the composition of reactor 2 and 10 and those of reactor 4 and 8 were found to be in close proximity on the lower right quadrant of the plot.
(PDF)

**S37 Fig.** Stacked barchat showing four *Thermoprotei* orders, relative abundances and their PCoA plot based on the Euclidean model (b). However, the nucleotide composition of reactor 3 and 8 were closely positioned in the lower left quadrant of the plot.
(PDF)

**S38 Fig.** Stacked barchat showing two *Thaurmarchaeota* order, the relative abundances (a) and their PCoA plot based on the Euclidean model (b). The plot revealed partial clustering of the nucleotide composition of reactor 6 and 9, positioned on the lower left quadrant of the plot, reactor 7 and 10, located near the y-axis, and reactor 2 and 11 located on the upper right quadrant of the plot. The nucleotide compositions of reactor 3 were in close proximity to those identified in reactor 7, positioned on the lower right quadrant of the plot. However, the other treatments' nucleotide compositions were found to be dissimilar.
(PDF)

**S39 Fig.** Stacked barchat showing five fungal phyla, the relative abundances (a) and their PCoA plots based on Euclidean model (b). The plot revealed dissimilarities of the nucleotide composition among the twelve studied treatments and were distributed within the four plot quadrant.
(PDF)

**S40 Fig.** Stacked barchat showing thirteen fungal classes, relative abundances (a) and their PCoA plot based on the Euclidean model (b). The PCoA plot revealed nucleotide composition dissimilarities among the twelve studied treatments.
(PDF)

**S41 Fig.** Stacked barchat showing 23 fungal orders, relative abundances (a) and their PCoA plot based on the Euclidean model (b). The nucleotide compositions of reactor 2 and 10, were positioned on the lower left quadrant; reactor 3 and 6, upper left quadrant; reactor 4 and 9, upper right quadrant; and reactor 8 and 11, on the lower right quadrant of the plot. All clustering partially in their respective quadrant.
(PDF)

**S42 Fig.** The stacked barchat showing the five *Ascomycota* classes, relative abaundances (a) and their PCoA plots based on the Euclidean model (b). The PCoA plots revealed dissimilarity of the nucleotide composition among the twelve treatments. However, the composition of reactor 11 was positioned singly on the lower left quadrant of the plot while those of reactor 10 were located along the y-axis (Negative PCoA 1 and Negative PCoA 2).
(PDF)

**S43 Fig.** The stacked barchat revealing eleven *Ascomycota* orders, relative abundances (a) and their PCoA plot based on the Euclidean model (b). The model revealed partial clustering of the identified nucleotide compositions of reactor 7 and 8 while those identified in reactor 2 and 8 and reactor 2 and 4 were found to reveal close proximity, located on the upper right quadrant of the plot. The *Ascomycota*'s nucleotide compositions of reactor 11 were singly positioned on the upper left quadrant of the plot.
(PDF)

**S44 Fig.** Stacked barchat showing three *Sordariomycetes* orders, relative abundances (a) and their PCoA plot based on the Euclidean model (b). The PCoA plots revealed partial clustering of the identified nucleotide reads in reactor 1 and 10 while other treatments revealed dissimilarities of the nucleotide composition.
(PDF)

**S45 Fig.** Stacked barchat showing the two *Eurotiomycetes* orders, the relative abundances (a) and their PCoA plot based on the Euclidean model (b). The nucleotide composition of reactor 1 and 3 clustered on the x-axis (positive PCoA 1 and Positive PCoA2); those of reactor 2 and 7, were positioned on the lower left quadrant while the communities of reactor 6 and 8, were found in close proximity on the lower right quadrant of the plot. However, the nucleotide compositions of reactor 11 were singly located on the upper right quadrant of the plot. (PDF)

**S46 Fig.** Stacked barchat showing four *Basidiomycota* classes, the relative abundances (a) and their PCoA plot based on the Euclidean model (b). The nucleotide composition of reactor 4 and 9 were found in close proximity, positioned on the upper right quadrant of the plot. The composition of reactor 3 and 11 were also closely located on the upper right quadrant, near the x-axis. The rest of the communities' nucleotide compositions in other treatments were distributed within the four plot quadrant. (PDF)

**S47 Fig.** Stacked barchat showing four *Basidiomycota* orders, relative abundances (a) and their PCoA plot based on the Euclidean model (b). The nucleotide compositions in the twelve treatments were dissimilar and were distributed within the four quadrants. (PDF)

**S48 Fig.** Stacked barchart showing the proportion of the *Agaricomycetes* order, relative abundances (a) and their PCoA plot based on the Euclidean model (b). The nucleotide composition affiliated to *Agaricomycete*s in reactor 1 and 3 were almost similar, positioned on the upper left quadrant of the plot. Their composition in reactor 7 and 9 were in close proximity, located in the lower left quadrant of the plot. The *Agaricomycete*s nucleotides in reactor 11 and 12 clustered partially on the lower left quadrant of the plot, those of reactor 5 were singly located in the upper right quadrant of the plot. (PDF)

**S49 Fig. Stacked barchat showing the two *Chytridiomycota* classes, and their relative abundances in the two treatments.** This is an indication of special substrates in the two treatments that were absent in other studied treatnents. (PDF)

**S50 Fig. Stacked barchat showing the four *Chytridiomycota* orders, and their relative abundances in the two treatments.** The orders are considered rare due to the fact that they were detected in only three treatments. (PDF)

**S51 Fig. Stacked barchat showing the three orders affiliated to unclassified fungal nucleotide reads.** The orders are considered rare due to the fact that they were detected in only three treatments. (PDF)

**S52 Fig.** The PCoA analysis revealing β-diversity of the twelve treatmenets at three taxa level: a) Domain, b) Phylum, and c) Class level. At the class level, the reads/nucleotide sequences of reactor 2 and 10 were partially similar, located in the lower left quadrant, those of reactor 8 and 11 were located on the upper right quadrant, reactor 4 and 9 compositions on the lower right quadrant, while those of reactor 3 and 6 were located on the lower left quadrant of the plot. (PDF)

## Acknowledgments

We appreciate Vincent Njunge, Collin Mutai, Tina Kyalo, Joyce Njoki Njuguna, Eunice Machuka, Muga Fredrick Nganga and Philis Emilda Ochieng of BecA-ILRI hub, Donald Otieno and Linnet Gohole of University of Eldoret and Lucyline Kajira Njeru of Moi University for their technical support.

## Author Contributions

**Conceptualization:** Samuel Mwangangi Muturi.

**Data curation:** Samuel Mwangangi Muturi.

**Formal analysis:** Samuel Mwangangi Muturi.

**Funding acquisition:** Samuel Mwangangi Muturi.

**Investigation:** Samuel Mwangangi Muturi.

**Methodology:** Samuel Mwangangi Muturi, Stephen Obol Opiyo.

**Project administration:** Samuel Mwangangi Muturi.

**Software:** Samuel Mwangangi Muturi.

**Validation:** Samuel Mwangangi Muturi.

**Visualization:** Samuel Mwangangi Muturi.

**Writing – original draft:** Samuel Mwangangi Muturi.

**Writing – review & editing:** Samuel Mwangangi Muturi, Lucy Wangui Muthui, Paul Mwangi Njogu, Justus Mong'are Onguso, Francis Nyamu Wachira, Stephen Obol Opiyo, Roger Pelle.

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
