## [Decision Letter · Decision Letter 0]

12 Oct 2020

PONE-D-20-20281

Metagenomics survey unravels diversities of biogas’ microbiomes with potential to enhance its’ productivity in Kenya

PLOS ONE

Dear Dr. Muturi,

Thank you for submitting your manuscript to PLOS ONE. After careful consideration, we feel that it has merit but does not fully meet PLOS ONE’s publication criteria as it currently stands. Therefore, we invite you to submit a revised version of the manuscript that addresses the points raised during the review process.

ACADEMIC EDITOR: Overall, the article can be improved further by providing a comparative study with the relevant literature on this topic.

We look forward to receiving your revised manuscript.

Kind regards,

Bawadi Abdullah

Academic Editor

PLOS ONE

Journal Requirements:

2. During your revisions, please note that a simple title correction is required: for grammatical accuracy, the title should read "Metagenomics survey unravels diversity of biogas microbiomes with potential to enhance productivity in Kenya". Please ensure this is updated in the manuscript file and the online submission information.

3. We note that you are reporting an analysis of a microarray, next-generation sequencing, or deep sequencing data set. PLOS requires that authors comply with field-specific standards for preparation, recording, and deposition of data in repositories appropriate to their field. Please upload these data to a stable, public repository (such as ArrayExpress, Gene Expression Omnibus (GEO), DNA Data Bank of Japan (DDBJ), NCBI GenBank, NCBI Sequence Read Archive, or EMBL Nucleotide Sequence Database (ENA)). In your revised cover letter, please provide the relevant accession numbers that may be used to access these data. For a full list of recommended repositories, see http://journals.plos.org/plosone/s/data-availability#loc-omics or http://journals.plos.org/plosone/s/data-availability#loc-sequencing.

4.Thank you for stating the following in the Funding Section of your manuscript:

[This work was supported in parts by the BecA-ILRI-hub through the Africa

Biosciences Challenge Funds (ABCF) fellowship Program and University of Eldoret through

the Annual research grants. The funders of ABCF fellowship had no role in experimental

design, data collection and processing, manuscript preparation and publishing.]

 [The funders had no role in study design, data collection and analysis, decision to publish, or preparation of the manuscript.]

We note that one or more of the authors is affiliated with the funding organization, indicating the funder may have had some role in the design, data collection, analysis or preparation of your manuscript for publication; in other words, the funder played an indirect role through the participation of the co-authors. If the funding organization did not play a role in the study design, data collection and analysis, decision to publish, or preparation of the manuscript and only provided financial support in the form of authors' salaries and/or research materials, please do the following:

Review your statements relating to the author contributions, and ensure you have specifically and accurately indicated the role(s) that these authors had in your study. These amendments should be made in the online form.

Confirm in your cover letter that you agree with the following statement, and we will change the online submission form on your behalf:

Reviewers' comments:

Reviewer's Responses to Questions

**Comments to the Author**

1. Is the manuscript technically sound, and do the data support the conclusions?

Reviewer #1: Yes

Reviewer #2: Yes

2. Has the statistical analysis been performed appropriately and rigorously? 

Reviewer #1: Yes

Reviewer #2: No

3. Have the authors made all data underlying the findings in their manuscript fully available?

Reviewer #1: Yes

Reviewer #2: Yes

4. Is the manuscript presented in an intelligible fashion and written in standard English?

Reviewer #1: Yes

Reviewer #2: Yes

5. Review Comments to the Author

Reviewer #1: The manuscript can be further improved based on the following comments:

1. Abstract- The abstract is too long and should be more concise to the important findings in the present work.

2. “Furthermore, 39.75% to 46.70% and 53.07% to 60.11% genes were conveyed to known and unknown proteins’- Perhaps to include the examples of known protein in the sentence.

3. “Hence, only four out of twelve treatments had high β-diversities, which could indicate genetic plasticity due to species evolution”- Is there any solid evidence for this statement?

4. Figure 1 & 2- Proper unit should be indicated in the Y-axis.

5. Figure 3- How about sections that are not indicating of any percentage? Any discussion for this?

6. “…….implying metabolic and genetic plasticity among the identified Clostridia species.”- More justifications are required for this statement.

7. Figure 4- Higher resolution is required.

8. “The local communities of the other treatments were observed to be significantly dissimilar”- why?

9. “Generally the reactor treatments comprised high salts concentrations….”- How high is the salt concentration?

10. “…low amino acid affinities among their identified species, which would suggest another selection pressure in the AD system”- Is there any evidence on this statement?

11. “Due to this co-operation, we further postulate the existence of a holobiontic mechanism in our treatments….” More evidences are required to support this statement.

12. How about the methodology to determine the reactor performance as indicated in Table 3?

Reviewer #2: The authors present interesting data on the biogas microbiom diversity over broad environmental variables and the efficiency of biogas reactor systems which offers insights into better management strategies that may boost biochemical limitations to successful biogas production. The authors infer that environmental and spatial variables affected primarily the biogas microbiome inhabitants and hence the production of biogas. The results show the significance of understanding the natural processes involved that contribute to certain defined variations. Overall the authors have described their findings within the acceptable level of fundamental clarity and depth. The manuscript is well written, has important industrial message, and should be of great interest to the readers. The results are well presented and the statistical analysis would be sufficient to support their arguments. Generally, it is a significant study and should be considered to be published in PLOS ONE.

6. PLOS authors have the option to publish the peer review history of their article (what does this mean?). If published, this will include your full peer review and any attached files.

Reviewer #1: No

Reviewer #2: No

---

## [Author Response · Author response to Decision Letter 0]

4 Dec 2020

A). ACADEMIC EDITOR COMMENTS 

1. Summary of the PLOS ONE's style requirements responses- including those for file naming 

i). PLOS guidelines for title page were followed as recommended. 

ii). Authors, and affiliations; abstract and introduction are ok, in that order. PLOS guidelines followed.

iii). “ABSTRACT” revised to “Abstract” 

iv). Level 1 headings: Bold type, 18pt font and in sentence case.

v). Level 2 headings: Bold type, 16pt font and in sentence case. 

vi). Guidelines for Figure citations and labeling followed as recommended.

vii). Guidelines for File and figure naming were also followed. 

viii). The documents is presented in double-space paragraph format. 

ix). Continuous line numbers inserted. 

x). PLOS Guidelines for formatting tables and table Citations in the text followed.

xi). Guidelines for Reference Citations in the text followed as recommended.

xii). PLOS Guidelines for Supporting Information Citations in the text followed.

xiii). Removed funding, and competing interests information from the acknowledgments section. 

xiv). References style revised as per the PLOS guidelines and identifiers added.

xv). Supporting information section, level 1 heading: Bold type font 18 pt. 

xvi). Supporting Information Captions formatted as per the PLOS guidelines.

xvii). Guidelines for file Naming for Supporting Information followed.

xviii). PACE used to prepare figures

xix). We chose to present our manuscripts as: i). Results ii). Discussion iii). Conclusion and iv). Materials and methods. As per the PLOS guidelines, the said elements can be presented in any order. 

xx). The order: Acknowledgements, supporting materials captions and References, was revised to Acknowledgements, References, Supporting information captions as recommended in the ending sections.

xxi). Figure captions are inserted immediately after the first paragraph in which the figure is cited. Figure files are uploaded separately. 

xxii). Tables are inserted immediately after the first paragraph in which they are cited.

xxiii). Supporting information files are uploaded separately. 

2. Title - Revised as recommended or stated by the Editor 

3. Datasets uploaded in a stable, public repository. The following are the accession no. mgm4776224.3; mgm4776218.3; mgm4776192.3; mgm4776124.3; mgm4775917.3; mgm4775884.3; mgm4775867.3; mgm4775851.3; mgm4775675.3; mgm4775550.3; mgm4775461.3; mgm4775281.3; 

4. Authors contribution removed/deleted from the manuscript 

I agree with the following statement. Note that most authors including those affiliated to the funding organization, joined the project at advanced level of the project.

“The funder provided support in the form of salaries and research materials for authors [SMM], but did not have any additional role in the study design, data collection and analysis, decision to publish, or preparation of the manuscript. The specific roles of each author are articulated in the ‘author contributions’ section.”

B). REVIEWER’S COMMENTS

1. Abstract- The abstract is too long and should be more concise to the important findings in the present work.

Response: We have revised and shortened the abstract, from 318 words to 295 words. 

2. “Furthermore, 39.75% to 46.70% and 53.07% to 60.11% genes were conveyed to known and unknown proteins’- Perhaps to include the examples of known protein in the sentence.

Response: Our focus is taxonomy of the identified microbes not their encoded proteins. The statistics state the percentages and nature (type of encoded proteins) of the genes and what was used in our taxonomic classifications or identification. We feel this is enough as far as the manuscript is concerned; else, we shall be entering into functionality of the identified microbes, contrary to the aim of the Manuscript. 

3. “Hence, only four out of twelve treatments had high β-diversities, which could indicate genetic plasticity due to species evolution”- Is there any solid evidence for this statement? 

Response: Yes, the PCoA plot were generated at (P≤ 0.05), and present a solid evidence for the above statement. We used nucleotide sequences (genetic composition) to determine β-diversity. High β-diversity is a function of evolution. Evolution leads to genetic versatility that may result to new species and strains. Also included citations; see Randall Hughes et al., 2008; Legendre P. and Caceres M. 2013.

4. Fig. 1 & 2-Proper unit should be indicated in the Y-axis.

Response: Revised to add more details on the Y-axis of the two figures and other figures.

5. Figure 3- How about sections that are not indicating of any percentage? Any discussion for this? 

Response: We’ve expand the Krona radial space filling display” to capture the comments and to reveal other un-visible percentages see Fig 3a & b.tiff. 

6. “…….implying metabolic and genetic plasticity among the identified Clostridia species.”- More justifications are required for this statement. 

Response: To the best of our knowledge, the statement is correct. The word is ‘implying’ which create an experimental gap that can be confirmed later. Also (Randall Hughes et al., 2008; Legendre P. and Caceres M. 2013). The statements is not stated as part of our findings but stated as part of our thought/suggestion based on literature.

7. Figure 4- Higher resolution is required.

Response: The resolution of the figure and other figures were improved through the PACE. 

8. “The local communities of the other treatments were observed to be significantly dissimilar”- why?

Response: Because all our analysis was conducted at P≤ 0.05. On the PCoA plot the “other treatments” were clearly separated at 95% confidence level or threshold. 

9. “Generally the reactor treatments comprised high salts concentrations….”- How high is the salt concentration? 

Response: Our systems had almost similar salt components and concentrations (unpublished) as those reported by Jha et al., 2013 and Langer et al., 2019. 

Furthermore, cow-dung treating reactors contain high amount of salts and ammonia (Usack J. and Angenent L. 2015; Gallert C., Bauer S. and Winter J., 1998; Fang C., Boe K. and Angelidaki I., 2011; Zhang et al., 2019; Rath K. and Rousk J., 2015; De Vrieze et al., 2007). 

10. “…low amino acid affinities among their identified species, which would suggest another selection pressure in the AD system”- Is there any evidence on this statement? 

Response: Yes based on the literature, low substrates lead to selection pressure, where more competitive species i.e those with high substrates affinities dominates the niche. The phenomenon has been previously reported in methanosarcina and methanosaeta species on acetate substrate [see Goux et al., 2015; Konopka et al., 2015 among others]. 

11. “Due to this co-operation, we further postulate the existence of a holobiontic mechanism in our treatments….” More evidences are required to support this statement.

Response: We have revised the statement to replace “that is used to regulate” with “which could be regulating”. Also have cited; Bispo PdC. et al., 2017; https://doi.org/10.1371/journal. pone.0188300. The term “Postulate” express our feeling on the synergism among the involved microbes based on our best knowledge concerning the biogas systems.

12. How about the methodology to determine the reactor performance as indicated in Table 3?

Response: The missing statement on the method and material to determine performance included. 

Others:

Also replaced “Geographic positioning system” with “Global positioning system” in the methodology section. The term is better than the previous one. 

Revised and inserted more institutions in the Ethics statements section.

Revised Bioinformatics and statistics section (see the revised Manuscript).

---

## [Editor Report · Decision Letter 1]

16 Dec 2020

Metagenomics survey unravels diversity of biogas microbiomes with potential to enhance productivity in Kenya

PONE-D-20-20281R1

Dear Dr. Muturi,

We’re pleased to inform you that your manuscript has been judged scientifically suitable for publication and will be formally accepted for publication once it meets all outstanding technical requirements.

Kind regards,

Bawadi Abdullah

Academic Editor

PLOS ONE
---

## [Editor Report · Acceptance letter]

23 Dec 2020

PONE-D-20-20281R1 

Metagenomics survey unravels diversity of biogas microbiomes with potential to enhance productivity in Kenya 

Dear Dr. Muturi:

I'm pleased to inform you that your manuscript has been deemed suitable for publication in PLOS ONE. Congratulations! Your manuscript is now with our production department. 

Kind regards, 

on behalf of

Dr. Bawadi Abdullah 

Academic Editor

PLOS ONE